# LLM-OAP: An LLM-based Data Augmentation Framework for Enhancing Order Acceptance Prediction in Mobility-on-Demand Systems

## Abstract

In Mobility-on-Demand (MoD) systems, drivers' order acceptance behaviour directly influences matching, pricing, and thus overall system efficiency. Traditional discrete choice models rely on pre-specified utility functions and error structures. This introduces specification risk and limits their ability to capture complex non-linear interactions, and correlated choices, reducing effectiveness for modelling driver decisions. Meanwhile, behavioural data often come from stated-preference (SP) surveys; these datasets are typically small-scale and based on hypothetical responses, which can be subjective and limit external validity, reducing predictive performance and generalisability. This paper proposes LLM-OAP, a novel framework that integrates large language model (LLM)-based data augmentation with machine learning (ML) to improve the estimation of drivers' order acceptance behaviour. Our method leverages an LLM to generate synthetic samples based on the real SP data and employs a curation scheme to mitigate implausibility, reduce bias, and maintain diversity. The augmented dataset is used to train ML models beyond fixed utility specifications. Evaluations on two types of SP datasets (covering full- and limited-information settings) show that our framework significantly enhances the performance of state-of-the-art ML models in order acceptance behaviour estimation, while maintaining good generalizability and explainability.

## 1 Introduction

Mobility-on-Demand (MoD) services such as ride-hailing and ride-sharing offer flexible access to transportation, with the potential to improve resource utilization and reduce congestion (Barbosa et al., 2018). Their operational performance, however, hinges on whether drivers accept the matched orders, because acceptance behaviour determines how effectively the platform matches riders to available vehicles. In practice, drivers operate as independent contractors and make choices based on personal preferences and constraints: they may accept or reject assigned orders, or actively seek requests they expect to be more profitable or convenient (Ashkrof et al., 2022; Urata et al., 2021). These behavioural decisions (i.e., choices) can disrupt supply-demand balance, affect service reliability, and complicate downstream optimisation.

Modeling and predicting driver behaviour is therefore essential for designing effective and efficient operational strategies like matching and pricing (Gao et al., 2022; Ricard & Bierlaire, 2025). Previous studies have approached this challenge using qualitative methods, e.g., focus group interviews (Ashkrof et al., 2020), and quantitative tools. e.g., discrete choice models (DCMs) (Ben-Akiva & Lerman, 1985; Train, 2009; Ashkrof et al., 2022). These models are typically estimated using data from stated preference (SP) surveys, where respondents are presented with hypothetical choice sets and asked to indicate their preferred option (Louviere et al., 2000; Ashkrof et al., 2022). This setup allows researchers to capture driver behavioural decisions under controlled conditions and explore the influence of specific attributes on choices.

However, DCMs rely on strong assumptions about the functional form of utility, typically linear relationships between observed variables and choices, which often fail to capture internal nonlinearities or more intrinsic interactions, leading to low prediction accuracy (Sifringer et al., 2020; Wang et al., 2021). They also inherit the independence of irrelevant alternatives (IIA) property of standard

models like the multinomial logit, which may not hold in ride-hailing contexts where alternatives often share similar attributes. In such cases, the unobserved components of utility across choices are correlated, violating the IIA assumption (Torres et al., 2011; Han et al., 2022; Sifringer et al., 2020).

Machine learning (ML) approaches can address these limitations by capturing complex, nonlinear behavioural patterns without relying on predefined utility functions. However, training strong ML predictors typically relies on large-scale, high-resolution behavioural datasets, which are rarely available in MoD. The reason is that conducting large SP surveys is manually costly and time consuming, and issues like respondent fatigue and hypothetical bias further limit their utility for data-hungry ML models (Tjuatja et al., 2024). Although ML models have the potential in learning rich behavioural patterns, their efficacy is practically constrained by data scarcity and potential distributional gaps between hypothetical SP responses and operational settings.

To that end, we propose LLM-OAP, an LLM-based data-augmentation framework for enhancing ML performance in order acceptance prediction. To be specific, we leverage an LLM to generate structured accept/reject order samples conditioned on feature-aware persona grouping and behavioural summaries, thereby enriching the original dataset's coverage and diversity. Then, a curation scheme is designed to further refine the synthetic data to improve its realism and validity. We train downstream ML predictors on the augmented data and evaluate on two types of SP datasets. LLM-OAP consistently improves predictive performance over strong baselines. Ablation studies further demonstrate that each key component of the framework contributes critically to these performance gains.

## 2 RELATED WORKS

This section reviews the relevant literature in two areas: (1) behaviour estimation in MoD, and (2) data augmentation techniques.

**Mobility Behaviour Estimation** Discrete choice models (DCMs) are widely used to estimate behaviours in MoD systems. Classical forms such as multinomial logit and probit relate observable attributes (e.g., trip distance, waiting time, dynamic pricing) to accept/reject decisions (Ben-Akiva & Lerman, 1985; Train, 2009). However, linear-in-parameters utilities and, for logit models, the independence of irrelevant alternatives (IIA) assumption can be restrictive when alternatives share latent components or interact nonlinearly, which limits prediction under correlation (Munizaga et al., 2000). Extensions (mixed/nested logit, latent-class models) relax IIA or add heterogeneity (Train, 2009), yet they still rely on strong utility and error-structure assumptions and struggle with high-dimensional or unstructured inputs. Tree-based methods (e.g., XGBoost (Chen & Guestrin, 2016)) and deep neural networks such as Tabular ResNet (Kadra et al., 2021) and Ensemble Hypernet (Mai et al., 2025) avoid fixed utility forms and can capture more intricate interactions. However, their performance is bounded by data: labels are scarce and imbalanced, and most datasets are small, hypothetical SP surveys.

**Traditional Data Augmentation** Oversampling and resampling techniques are two commonly used techniques in traditional data augmentation, such as SMOTE (Chawla et al., 2002), ADASYN (He et al., 2008), MGVAE (Ai et al., 2023), and LITO (Yang et al., 2024), which balance class distributions by interpolating between minority samples. Other approaches rely on feature engineering and perturbation, including noise injection or attribute swapping, to increase data variability without altering the underlying label distribution. These methods are computationally efficient and easy to implement, but they often fail to capture complex feature dependencies in high-dimensional datasets.

On the other hand, generative models learn the joint distribution of features to produce synthetic rows, and often deliver better performance than sampling techniques. Recent tabular generators include CTGAN (Xu et al., 2019), CTAB-GAN (Zhao et al., 2021), TabGAn (Ashrapov, 2020), and OCTGAN (Kim et al., 2021), as well as diffusion-based methods such as TabDDPM (Kotelnikov et al., 2023), TabDiff (Shi et al., 2024), and CausalDiffTab (Zhang et al., 2025), which have advanced fidelity for continuous features and mixed data. While these methods have achieved progress in modeling continuous attributes, they typically require extensive fine-tuning and hyperparameter optimization to remain stable. Moreover, their performance often degrades when handling categorical variables or improving the representation of minority classes, limiting their effectiveness in real-world applications.

**LLMs for Data Augmentation** Large language models (LLMs) have shown great promise in synthetic data generation (Brand et al., 2023; Mirchandani et al., 2023; Gruver et al., 2023; Xu et al., 2024). Unlike traditional data augmentation approaches, LLMs can be guided through prompt-based in-context learning to produce structured data that incorporates semantic knowledge. For instance, EPIC employs CSV-style prompts, class balancing, and variable mapping to generate high-fidelity synthetic tables (Kim et al., 2024). CuratedLLM highlights the synergy of generation and curation, achieving strong performance in ultra low-data regimes (Seedat et al., 2023). Pred-LLM (Nguyen et al., 2024) further demonstrates the feasibility of LLMs to produce coherent tabular structures across diverse domains. In mobility choice modeling, Tzachristas et al. (2025) employ personalized prompts to simulate behavioural preferences at scale, improving realism in synthetic surveys but still facing challenges in feature-rich datasets.

Despite these advances, current LLM-based augmentation methods face several limitations. First, most rely on static prompt designs that do not adapt to heterogeneous feature spaces, making them less effective when handling high-dimensional or imbalanced tabular datasets. Second, quality control is often limited to one-off post-processing strategies (e.g., similarity checks, re-ranking), which cannot systematically correct generation biases. Moreover, without mechanisms such as feature-aware grouping or behaviour summarization, generated samples may lack alignment with underlying behavioural patterns. Finally, insufficient context diversity often leads to repetitive or logically inconsistent outputs, including hallucinations (Han et al., 2024; Peykani et al., 2025). In this work, we propose LLM-OAP, an LLM-based augmentation framework that combines feature-aware persona grouping, behaviour-informed generation with confidence- and uncertainty-based curation, thereby overcoming the limitations of static prompting and one-off post-processing strategies while improving fidelity and robustness of synthetic data.

## 3 METHODOLOGY

In this section, we first provide an overview of the proposed LLM-OAP framework, and then describe how an LLM is leveraged to perform group-based data augmentation. Next, we introduce the designed curation scheme based on confidence and uncertainty evaluation to filter low-quality samples, yielding an augmented dataset that is reliable and effective for training downstream models.

### 3.1 OVERALL FRAMEWORK

We use a cross-sectional stated-preference dataset of ride-hailing drivers from the United States and the Netherlands (Ashkrof et al., 2022). The dataset contains roughly 3,000 order acceptance records from hundreds of drivers across several cities, each with about 50 features spanning driver attributes (age, working hours, historical acceptance rate, education) and order context (location, pick-up distance, estimated fare, waiting time, dynamic tip). Variables are a heterogeneous mix of continuous and categorical types. The details of this dataset are provided in Appendix A.1.

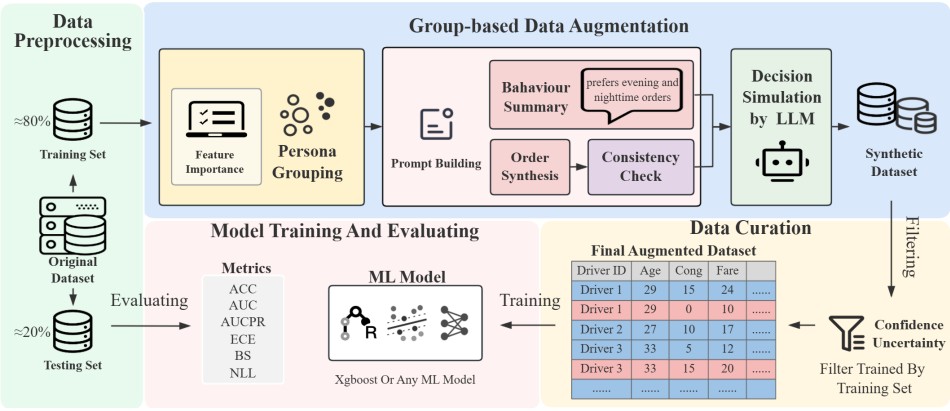

Figure 1: The illustration of LLM-OAP framework.

Given the dataset of drivers' order acceptance, the overall framework (shown in Figure 1) proceeds as follows. First, the raw dataset is divided into training and testing sets, with the testing set preserved for final evaluation and the training set used for augmentation. Second, the LLM performs group-based data augmentation through four sequential steps: *feature-aware persona grouping*, *behaviour summary and order synthesis within each group*, *consistency check*, and *decision simulation*. These steps progressively generate synthetic dataset. The synthetic dataset is then evaluated in confidence and uncertainty by a filter trained with the original training set, and low-quality samples are discarded to further improve the overall quality of the dataset. Finally, we produce an augmented dataset characterized by both high fidelity and diversity, and use it to train ML models such as XGBoost for order acceptance prediction.

## 3.2 Group-based Data Augmentation

Before augmentation, we preprocess the data to optimize feature organization, enabling more effective utilization by the LLM. Let the original feature set be $\mathcal{F} = \{f_1, \ldots, f_d\}$, where $d$ is the number of features. We partition features into environmental features $\mathcal{F}_{\text{env}}$ (e.g., spatiotemporal state, passenger attributes, and order properties) and individual-specific features $\mathcal{F}_{\text{ind}}$ (e.g., driver age, cumulative working hours, part-time status), with $\mathcal{F}_{\text{env}} \cap \mathcal{F}_{\text{ind}} = \emptyset$. We reorder the records in the dataset according to the feature values in $\mathcal{F}_{\text{env}}$ and $\mathcal{F}_{\text{ind}}$, such that $\mathcal{F}_{\text{env}}$ precede $\mathcal{F}_{\text{ind}}$ (see Appendix A.1 for more details of reordering.). In addition, the irrelevant features (e.g., Block: the survey block identifier) are removed, since they neither reflect drivers' behavioural patterns nor provide meaningful contextual information. The resulting reduced feature set is denoted as $\mathcal{F}' \subseteq \mathcal{F}$ and the final preprocessed dataset as $\mathcal{D} = \{(x_i, y_i)\}_{i=1}^N$, where $x_i \in \mathbb{R}^{d'}$ with $d' = |\mathcal{F}'|$ and $y_i \in \{0, 1\}$ indicates acceptance (1) or rejection (0). Building upon the organized dataset, we propose a group-based augmentation strategy that progressively generates high-quality synthetic records through four sequential steps: *feature-aware persona grouping*, *behaviour summary and order synthesis*, *consistency check*, and *decision simulation*.

**Feature-aware Persona Grouping** To capture the heterogeneity in drivers' order acceptance preference, we first compute permutation feature importance scores $\text{I}(f_j)$ for each feature $f_j \in \mathcal{F}'$ (A detailed description of feature importance is given in Appendix A.2.). After that, we construct a small set of four categorical features:

$$\mathcal{F}^* = \{f_{j_1}, f_{j_2}, f_{j_3}\} \cup \{g_{\text{age}}\}$$

where $g_{\text{age}} = \text{bin}(\text{age})$ is an additional feature column we manually created. $\text{bin}(\cdot)$ means grouping drivers' ages (ranging from 20 to 90 in the dataset) into 10-year bins. The remaining three features $\{f_{j_1}, f_{j_2}, f_{j_3}\} \subseteq \mathcal{F}'$ are randomly selected from the top 10 features ranked by $\text{I}(f_j)$. The number of features in $\mathcal{F}^*$ balances their importance (i.e., effects in the order acceptance) and the granularity of grouping. For example, including more features reduces the number of records within each group, potentially diminishing the quality of the records generated by the LLM. In this paper, we randomly select Beginners (indicator of drivers with less than 12 months of experience), NY_CA (indicators of locations, New York or California), and Part (indicator of part-time driver) as the features, and the detailed rationale behind this selection is provided in Appendix A.9.

Given the selected categorical features, each driver's records are clustered into a persona group $G_k$ based on these features. Formally, the dataset $\mathcal{D}$ is partitioned into $K = 32$ disjoint persona groups:

$$\mathcal{D} = \bigcup_{k=1}^K G_k, \quad G_k \cap G_{k'} = \emptyset \quad \forall k, k' \in \{1, \ldots, K\}, k \neq k'.$$

where $K$ is determined by the Cartesian product of category values across $\mathcal{F}^*$, such that each group corresponds to a unique combination of categorical attributes. Groups with no assigned records are discarded, ensuring that only valid persona groups are retained. For each persona group $G_k$, we compute the acceptance rate

$$r_k = \frac{1}{|G_k|} \sum_{(x_i, y_i) \in G_k} \mathbf{1}(y_i = 1),$$

and rejection rate $1 - r_k$, which together with persona attributes $\mathcal{F}_k^*$, construct the prompt to guide the subsequent augmentation process.

**Behaviour Summary and Order Synthesis**   For each persona group $G_k$, we provide the LLM with: (1) the group-level acceptance/rejection rates $(r_k, 1 - r_k)$; (2) persona attributes $\mathcal{F}_k^*$; (3) a balanced set of examples $\mathcal{H}_k = \{(x_i, y_i)\}$, which is randomly drawn from the current persona group, where the number of accepted and rejected records are approximately equal, i.e., $|\{y_i = 1\}| \approx |\{y_i = 0\}|$. Through contextual learning, we prompt the LLM to generate natural language summaries that capture the behavioural tendencies of a group (e.g., "more likely to accept short-distance orders with low waiting times" or "prefers evening and nighttime orders and responds more positively to peak-hour requests"). At the same time, owing to the complexity and heterogeneous constraints of the features such as driver age, cumulative working hours, and guaranteed tips, we avoid using the LLM completion mode (i.e., provide LLM with some examples and ask it to automatically continue writing based on the examples without additional instructions), which risks producing implausible values due to limited example coverage. Instead, we adopt traditional explicit instructions, such as "generate $n$ reasonable, realistic ride-hailing orders without labels based on the above information", to guide the LLM to synthesize new order records $\tilde{\mathcal{X}}_k$ for each group $k$, without labels, i.e., acceptance (1) or rejection (0):

$$\tilde{\mathcal{X}}_k = \{\tilde{x}_1, \ldots, \tilde{x}_m\}, \quad \tilde{x}_j \in \mathbb{R}^{|\mathcal{F}'|}.$$

Such explicit prompting helps avoid infeasible attribute combinations (e.g., long-distance trips paired with low fares). The complete prompts are provided in Appendix A.4.

**Consistency Check**   To ensure the logical validity and structural consistency of the generated records, we design an ID-attribute consistency check mechanism to mitigate the inherent instability of LLM outputs. Specifically, let $\mathcal{I}_k = \{(id, a_{id})\}$ denote the original set of driver IDs with their individual-specific attributes (i.e., features) in group $G_k$. For each synthesized record $(\tilde{id}, \tilde{a}_{id}) \in \tilde{\mathcal{X}}_k$, if $\tilde{id} \notin \mathcal{I}_k$, the record is discarded; if $\tilde{id} \in \mathcal{I}_k$ but $\tilde{a}_{id} \neq a_{\tilde{id}}$, then we update the $\tilde{a}_{id}$ by replacing every mismatched individual-specific attribute in $\tilde{a}_{id}$ with the corresponding value in $a_{\tilde{id}}$.

We predefine a total target number of synthetic records for the training set and the number of records generated in each group should be proportionally to its record ratio among the training set. We count the number of valid synthetic records. If it falls short of the target number for a group, we repeat the augmentation steps until the required number of records are generated.

**Decision Simulation**   Finally, we provide the LLM with both the persona summaries and the synthesized records (without labels), prompting it to simulate acceptance/rejection decisions that are consistent with the persona's behavioural tendencies for each group. The simulated decisions (i.e., labels) are parsed from the natural language, integrated with attributes, and structured into the training set. By the group-based data augmentation, the synthesized records preserve consistency with both attributes and group-level behavioural tendencies.

## 3.3   DATA CURATION

Although the group-based data augmentation process generates diverse samples, not all of them are of sufficient quality. To address this issue, we introduce a data curation scheme for further improving the synthesis. Specifically, we train a filtering model $p_\theta$ (XGBoost in this paper) on the original training set, which is subsequently adopted to estimate both the confidence and uncertainty of each synthetic sample (Kwon et al., 2020; Seedat et al., 2022). For each synthetic sample $(\tilde{x}, \tilde{y})$, the confidence is defined as:

$$\text{Conf}(\tilde{x}) = \frac{1}{E} \sum_{e=1}^{E} p_\theta^{(e)}(\tilde{y} \mid \tilde{x}), \tag{1}$$

where $E$ denotes the number of checkpoints obtained during the training of the filtering model , and $p_\theta^{(e)}(\tilde{y} \mid \tilde{x})$ represents the probability assigned to label $\tilde{y}$ when predicting sample $\tilde{x}$ under the model $p_\theta^{(e)}$ at checkpoint $e$. A higher confidence value indicates that the model assigns higher certainty to its prediction. The uncertainty is estimated by averaging the variance of the derived probabilities across checkpoints:

$$\text{Unc}(\tilde{x}) = \frac{1}{E} \sum_{e=1}^{E} \left( p_\theta^{(e)}(\tilde{y} \mid \tilde{x}) \cdot (1 - p_\theta^{(e)}(\tilde{y} \mid \tilde{x})) \right), \tag{2}$$

which reflects the degree of inconsistency or variability in the model's predictions for a sample. A higher value indicates that the model is less certain about the assigned label.

With the confidence and uncertainty, synthetic samples are retained only if they satisfy the condition:

$$\text{Conf}(\tilde{x}) \geq \tau_c \quad \wedge \quad \text{Unc}(\tilde{x}) \leq \tau_u, \tag{3}$$

where $\tau_c$ and $\tau_u$ denote the thresholds for confidence and uncertainty, respectively. Using the above condition, we generally keep high-confidence and low-uncertainty samples. We also preserve a small fraction of samples a high uncertainty to enhance data diversity.

The proposed data curation ensures that only reliable and representative LLM-generated samples are preserved, thereby improving the overall fidelity of the final augmented dataset.

## 4 EXPERIMENTS

### 4.1 SETUP

Our framework employs the QWEN3-Plus model as the LLM component (Performance under other LLMs is detailed in section 4.4.), we retain all default settings and only modify the key parameter temperature = 0.3, while top-p and top-k remain at their default values. All experiments were conducted using three different random seeds (2025, 42, 0), with results averaged to ensure robustness. Computations were carried out on a laptop configured with an AMD Ryzen 9 7845HX 3.00GHz CPU, 64GB of RAM, and an NVIDIA GeForce RTX 4060 laptop-grade GPU.

**Datasets** Following Ashkrof et al. Ashkrof et al. (2022), we use two platform information-sharing settings for the same order context for predicting drivers' order acceptance decisions. Under *Baseline Information Provision (BIP)*, drivers decide to accept or reject an order using only the information currently provided by the platform, e.g., their current spatiotemporal status, passenger characteristics, order rating and surge pricing. Under *Additional Information Provision (AIP)*, the platform reveals extra information for the same order, e.g., estimated trip fare, guaranteed tips and estimated delay, giving drivers a second opportunity to decide. We construct two datasets aligned with these scenarios and evaluate our method on both.

The two settings represent distinct and practically relevant decision conditions. AIP provides a *high information setting* that includes monetary and operational attributes; it shows what our approach can achieve when extra features are available. BIP removes those added attributes and is therefore a *reduced or limited information setting*; it tests whether our approach still improves predictive performance when the feature space is restricted. Because AIP contains more features, we report results for AIP first to establish a high-information reference, and then report BIP to assess robustness when some important features are withheld.

**Evaluation Metrics** We evaluate the proposed method on two dimensions: classification performance and the uncertainty quantification quality. To account for class imbalance between "accept" and "reject" decisions, we report overall accuracy (ACC), Area Under the ROC Curve (AUC), and Area Under the Precision–Recall Curve (AUCPR). These metrics capture predictive performance across thresholds, with AUCPR being particularly informative under imbalance. We further assess the quality of the predicted probabilities using Expected Calibration Error (ECE), the Brier Score (BS), and Negative Log-Likelihood (NLL) Gneiting & Raftery (2007).

To validate the stability of our approach and the reliability and significance of the performance improvements, we also conducted stability analysis and significance testing, which are presented in appendix A.6.

**Baseline Models** Baseline models include a linear model: logistic regression (LR), and nonlinear models, including decision trees (DT), Tabular ResNet (TabResNet) (Kadra et al., 2021), XGBoost (Chen & Guestrin, 2016), Ensemble Hypernet (Ens_Hyper) (Mai et al., 2025), and Support Vector Machine (SVM). Among these models, we selected the best-performing ones as backbone models, and conducted comparisons with Guided Persona-based AI Surveys (GPAIS) (Tzachristas et al., 2025), CTAB-GAN (Zhao et al., 2021), TabDDPM (Kotelnikov et al., 2023), Curated LLM (CLLM) (Seedat et al., 2023) and Pred-LLM (Nguyen et al., 2024).

Table 1: Performance comparison of different models on AIP scenario. "Backbone w/ Method" means the backbone is trained on real data augmented with synthetic samples generated by *Method*.

| Model | ACC ↑ | AUC ↑ | AUCPR ↑ | ECE ↓ | BS ↓ | NLL ↓ |
|---|---|---|---|---|---|---|
| LR | 0.7991 | 0.6357 | 0.2974 | **0.0359** | 0.1552 | 0.4844 |
| DT | 0.8121 | 0.7057 | 0.4295 | 0.0494 | 0.1414 | 0.4945 |
| TabResNet | 0.7929 | 0.7379 | 0.4644 | 0.1038 | 0.1546 | 0.4915 |
| Ens_Hyper | 0.8102 | 0.7355 | 0.4439 | 0.0413 | 0.1404 | 0.4467 |
| SVM | **0.8246** | 0.7214 | 0.4547 | 0.0363 | 0.1369 | 0.4365 |
| XGBoost | 0.8146 | **0.7875** | **0.5305** | 0.0466 | **0.1289** | **0.4060** |
| TabResNet w/ GPAIS | 0.7683 | 0.6927 | 0.3915 | 0.1061 | 0.1662 | 0.5324 |
| TabResNet w/ CTAB-GAN | 0.7962 | 0.7458 | 0.4726 | 0.0662 | 0.1456 | 0.4661 |
| TabResNet w/ TabDDPM | 0.8073 | 0.7435 | 0.5065 | 0.0914 | 0.1445 | 0.4735 |
| TabResNet w/ CLLM | 0.8030 | 0.7489 | 0.5016 | 0.1001 | 0.1488 | 0.5006 |
| TabResNet w/ Pred-LLM | 0.8054 | 0.7492 | 0.4833 | 0.0886 | 0.1456 | 0.4759 |
| **TabResNet w/ LLM-OAP** | **0.8237** | **0.7765** | **0.5323** | **0.0475** | **0.1310** | **0.4245** |
| Ens_Hyper w/ GPAIS | 0.7669 | 0.7135 | 0.3843 | 0.0792 | 0.1615 | 0.5026 |
| Ens_Hyper w/ CTAB-GAN | 0.8112 | 0.7568 | 0.4915 | 0.0714 | 0.1394 | 0.4544 |
| Ens_Hyper w/ TabDDPM | 0.8096 | 0.7514 | 0.5018 | 0.0731 | 0.1396 | 0.4597 |
| Ens_Hyper w/ CLLM | 0.8064 | 0.7491 | 0.5070 | 0.0775 | 0.1408 | 0.4611 |
| Ens_Hyper w/ Pred-LLM | 0.8146 | 0.7614 | 0.4962 | 0.0541 | 0.1363 | 0.4309 |
| **Ens_Hyper w/ LLM-OAP** | **0.8266** | **0.7889** | **0.5435** | **0.0408** | **0.1278** | **0.4105** |
| SVM w/ GPAIS | 0.7649 | 0.6779 | 0.3348 | 0.1207 | 0.1660 | 0.5099 |
| SVM w/ CTAB-GAN | 0.8285 | 0.7516 | 0.5144 | 0.0420 | 0.1337 | 0.4211 |
| SVM w/ TabDDPM | 0.8232 | 0.7454 | 0.5069 | 0.0464 | 0.1334 | 0.4307 |
| SVM w/ CLLM | 0.8285 | 0.7455 | 0.4861 | 0.0463 | 0.1398 | 0.4455 |
| SVM w/ Pred-LLM | 0.8263 | 0.7534 | 0.5079 | 0.0409 | 0.1329 | 0.4276 |
| **SVM w/ LLM-OAP** | **0.8372** | **0.7745** | **0.5436** | **0.0309** | **0.1256** | **0.4065** |
| XGBoost w/ GPAIS | 0.8006 | 0.7282 | 0.4471 | 0.0523 | 0.1438 | 0.4487 |
| XGBoost w/ CTAB-GAN | 0.8276 | 0.7940 | 0.5435 | 0.0425 | 0.1269 | 0.4028 |
| XGBoost w/ TabDDPM | 0.8296 | 0.7967 | 0.5630 | 0.0413 | 0.1230 | 0.4027 |
| XGBoost w/ CLLM | 0.8309 | 0.7907 | 0.5484 | 0.0379 | 0.1261 | 0.4032 |
| XGBoost w/ Pred-LLM | 0.8304 | 0.7964 | 0.5605 | 0.0410 | 0.1246 | 0.3973 |
| **XGBoost w/ LLM-OAP** | **0.8396** | **0.8198** | **0.5894** | **0.0319** | **0.1194** | **0.3834** |

## 4.2 DATASET AIP: HIGH-INFORMATION SETTING

**Comparative Study** We begin by evaluating LLM-OAP under dataset AIP, which provides richer information for each order. This high-information setting provides a strong basis for assessing the model's predictive performance. As shown in Table 1, we evaluate LLM-OAP against serveral augmentation baselines, including generative approach (CTAB-GAN), diffusion-based method (TabD-DPM), and LLM-based methods (GPAIS, CLLM, and Pred-LLM), across four backbone models (TabResNet, Ens_Hyper, SVM and XGBoost). Each method generates 3,000 synthetic samples. Compared to the strongest LLM-based baseline, such as Pred-LLM, LLM-OAP improves AUCPR by 0.0289 on XGBoost. The quality of probabilistic predictions also improves, with consistent reductions in ECE, BS, and NLL across all backbones. In fact, LLM-OAP outperforms *all* baselines across all six evaluation metrics. These results indicate that LLM-OAP produces higher-quality synthetic data and delivers stronger downstream gains.

**Ablation Study** To systematically quantify the contribution of each component of our method LLM-OAP, we conduct ablation studies by isolating the effects of feature importance, persona grouping, consistency check across four scenarios: **S1**: persona grouping is performed without feature importance; **S2**: grouping is entirely removed; **S3**: the consistency check is omitted; **S4**: the full version of LLM-OAP.

As shown in Table 2, performance degrades noticeably when either feature-aware grouping or the consistency check is removed, with S2 showing the most pronounced drop due to the loss of behavioural heterogeneity modeling. S1 performs better than S2 but still lags behind S4, indicating that feature importance is crucial for forming balanced and informative persona groups. Similarly, excluding the consistency check (S3) reduces sample reliability, leading to less stable results. By contrast, the full framework (S4) consistently achieves the strongest outcomes across all metrics and models, confirming that each component contributes meaningfully to the effectiveness of LLM-OAP. Furthermore, we investigate the role of the curation scheme by varying the proportion of curated samples retained. As shown in Figure 2(a), using all generated data without curation yields suboptimal performance, while overly aggressive filtering reduces data diversity. Retaining around

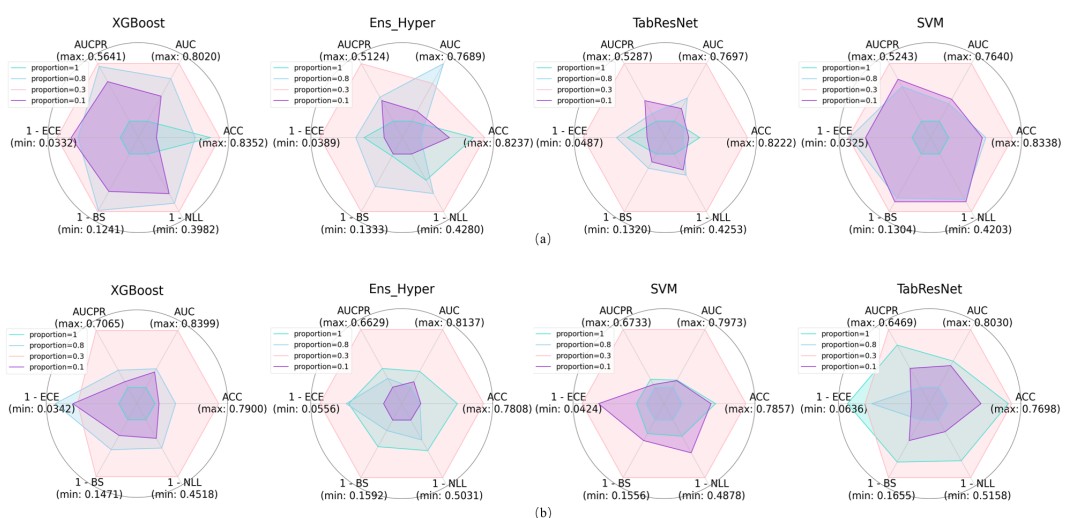

Figure 2: Performance of various models with different curation proportions on AIP (a) and BIP (b).

30% of high-confidence samples consistently achieves the best trade-off between fidelity and diversity, leading to the strongest performance across metrics. To further enhance dataset diversity, we additionally preserve a small portion of high-uncertainty samples (around 10%), as detailed in Appendix A.8.

Table 2: Ablation Studies on AIP scenario. S1: grouping without feature importance, S2: without grouping, S3: without consistency check, S4: full version of LLM-OAP

| Study | Models | ACC ↑ | AUC ↑ | AUCPR ↑ | ECE ↓ | BS ↓ | NLL ↓ |
|---|---|---|---|---|---|---|---|
| S1 | TabResNet | 0.8169 | 0.7517 | 0.5098 | 0.0581 | 0.1377 | 0.4384 |
| | Ens_Hyper | 0.8213 | 0.7621 | 0.5090 | 0.0603 | 0.1363 | 0.4386 |
| | SVM | 0.8266 | 0.7490 | 0.5155 | 0.0602 | 0.1328 | 0.4333 |
| | XGBoost | 0.8362 | 0.7955 | 0.5545 | 0.0432 | 0.1244 | 0.4015 |
| S2 | TabResNet | 0.8015 | 0.7552 | 0.4761 | 0.0683 | 0.1450 | 0.4561 |
| | Ens_Hyper | 0.8044 | 0.7674 | 0.4883 | 0.0658 | 0.1403 | 0.4387 |
| | SVM | 0.8030 | 0.7548 | 0.4787 | 0.0603 | 0.1396 | 0.4386 |
| | XGBoost | 0.8141 | 0.7891 | 0.5235 | 0.0435 | 0.1294 | 0.4051 |
| S3 | TabResNet | 0.8169 | 0.7517 | 0.5098 | 0.0581 | 0.1377 | 0.4384 |
| | Ens_Hyper | 0.8213 | 0.7621 | 0.5090 | 0.0603 | 0.1363 | 0.4386 |
| | SVM | 0.8266 | 0.7490 | 0.5155 | 0.0602 | 0.1328 | 0.4333 |
| | XGBoost | 0.8362 | 0.7955 | 0.5545 | 0.0432 | 0.1244 | 0.4015 |
| S4 | TabResNet | **0.8237** | **0.7765** | **0.5323** | **0.0475** | **0.1310** | **0.4245** |
| | Ens_Hyper | **0.8266** | **0.7889** | **0.5435** | **0.0408** | **0.1278** | **0.4105** |
| | SVM | **0.8372** | **0.7745** | **0.5436** | **0.0309** | **0.1256** | **0.4065** |
| | XGBoost | **0.8396** | **0.8198** | **0.5894** | **0.0319** | **0.1194** | **0.3834** |

## 4.3 DATASET BIP: LIMITED-INFORMATION SETTING

**Comparative study** Compared to the AIP, which provides complete contextual information, the BIP setting omits critical features such as trip fare, guaranteed tip, and traffic congestion, substantially increasing the difficulty of learning drivers' order acceptance preferences. This limited-information setting presents a more challenging and realistic scenario, testing whether LLM-OAP can maintain performance when important features are unavailable. As shown in Table 3, LLM-OAP achieves the best or second-best performance across models and metrics in the more challenging BIP setting. Compared with other methods, our approach yields larger gains, particularly in ECE, BS, and NLL, confirming its sharper probability estimates under partial information.

**Ablation Study** The ablation setup follows the same design as the AIP, where we selectively remove different components of LLM-OAP to assess their contributions. As shown in Table 4, removing

Table 3: Performance comparison of different models on BIP scenario.

| Model | ACC ↑ | AUC ↑ | AUCPR ↑ | ECE ↓ | BS ↓ | NLL ↓ |
|---|---|---|---|---|---|---|
| LR | 0.7086 | 0.6996 | 0.5062 | **0.0370** | 0.1889 | 0.5613 |
| DT | 0.7293 | 0.7321 | 0.5439 | 0.0573 | 0.1822 | 0.5927 |
| TabResNet | 0.7432 | 0.7525 | 0.5866 | 0.1159 | 0.1883 | 0.6100 |
| Ens_Hyper | 0.7466 | 0.7644 | 0.5972 | 0.0594 | 0.1721 | 0.5203 |
| SVM | 0.7567 | 0.7521 | 0.6017 | 0.0442 | 0.1714 | 0.5211 |
| XGBoost | **0.7678** | **0.8034** | **0.6353** | 0.0501 | **0.1607** | **0.4919** |
| TabResNet w/ GPAIS | 0.6927 | 0.6710 | 0.4712 | 0.1220 | 0.2155 | 0.6500 |
| TabResNet w/ CTAB-GAN | 0.7433 | 0.7551 | 0.5864 | 0.0927 | 0.1833 | 0.5754 |
| TabResNet w/ TabDDPM | 0.7548 | 0.7633 | 0.5919 | 0.1027 | 0.1821 | 0.5882 |
| TabResNet w/ CLLM | 0.7562 | 0.7807 | 0.6074 | 0.1214 | 0.1873 | 0.6010 |
| TabResNet w/ Pred-LLM | 0.7600 | 0.7824 | 0.6062 | 0.1089 | 0.1823 | 0.5872 |
| **TabResNet w/ LLM-OAP** | **0.7698** | **0.8030** | **0.6469** | **0.0697** | **0.1655** | **0.5158** |
| Ens_Hyper w/ GPAIS | 0.6999 | 0.6832 | 0.4735 | 0.1030 | 0.2095 | 0.6314 |
| Ens_Hyper w/ CTAB-GAN | 0.7563 | 0.7738 | 0.6064 | 0.0775 | 0.1741 | 0.5409 |
| Ens_Hyper w/ TabDDPM | 0.7519 | 0.7653 | 0.5920 | 0.1026 | 0.1803 | 0.5666 |
| Ens_Hyper w/ CLLM | 0.7544 | 0.7818 | 0.6090 | **0.0510** | 0.1735 | 0.5232 |
| Ens_Hyper w/ Pred-LLM | 0.7677 | 0.7862 | 0.6108 | 0.0596 | 0.1729 | 0.5239 |
| **Ens_Hyper w/ LLM-OAP** | **0.7808** | **0.8137** | **0.6629** | 0.0556 | **0.1592** | **0.5031** |
| SVM w/ GPAIS | 0.6980 | 0.6881 | 0.4924 | 0.0610 | 0.1916 | 0.5657 |
| SVM w/ CTAB-GAN | 0.7646 | 0.7570 | 0.6086 | 0.0577 | 0.1687 | 0.5209 |
| SVM w/ TabDDPM | 0.7625 | 0.7608 | 0.6165 | 0.0528 | 0.1703 | 0.5186 |
| SVM w/ CLLM | 0.7720 | 0.7747 | 0.6143 | 0.0517 | 0.1724 | 0.5245 |
| SVM w/ Pred-LLM | 0.7717 | 0.7794 | 0.6145 | 0.0515 | 0.1705 | 0.5205 |
| **SVM w/ LLM-OAP** | **0.7857** | **0.7973** | **0.6733** | **0.0424** | **0.1556** | **0.4878** |
| XGBoost w/ GPAIS | 0.7081 | 0.7074 | 0.5020 | 0.0535 | 0.1913 | 0.5640 |
| XGBoost w/ CTAB-GAN | 0.7670 | 0.8091 | 0.6610 | 0.0481 | 0.1543 | 0.4770 |
| XGBoost w/ TabDDPM | 0.7741 | 0.8122 | 0.6638 | 0.0485 | 0.1559 | 0.4741 |
| XGBoost w/ CLLM | 0.7682 | 0.8087 | 0.6579 | 0.0524 | 0.1590 | 0.4832 |
| XGBoost w/ Pred-LLM | 0.7759 | 0.8185 | 0.6710 | 0.0469 | 0.1578 | 0.4788 |
| **XGBoost w/ LLM-OAP** | **0.7900** | **0.8399** | **0.7065** | **0.0389** | **0.1471** | **0.4518** |

Table 4: Ablation Studies on BIP scenario.

| Study | Models | ACC ↑ | AUC ↑ | AUCPR ↑ | ECE ↓ | BS ↓ | NLL ↓ |
|---|---|---|---|---|---|---|---|
| S1 | TabResNet | 0.7644 | 0.7769 | 0.6149 | 0.0814 | 0.1692 | 0.5195 |
| | Ens_Hyper | 0.7630 | 0.7780 | 0.6169 | **0.0553** | 0.1692 | 0.5200 |
| | SVM | 0.7760 | 0.7678 | 0.6331 | 0.0472 | 0.1641 | 0.5052 |
| | XGBoost | 0.7736 | 0.8190 | 0.6749 | 0.0413 | 0.1536 | 0.4717 |
| S2 | TabResNet | 0.7293 | 0.7581 | 0.5569 | 0.0870 | 0.1848 | 0.5582 |
| | Ens_Hyper | 0.7351 | 0.7775 | 0.5919 | 0.0815 | 0.1778 | 0.5341 |
| | SVM | 0.7370 | 0.7737 | 0.5749 | 0.0629 | 0.1739 | 0.5218 |
| | XGBoost | 0.7428 | 0.7888 | 0.6160 | 0.0478 | 0.1660 | 0.4951 |
| S3 | TabResNet | 0.7692 | 0.7985 | 0.6348 | 0.0794 | 0.1759 | 0.5289 |
| | Ens_Hyper | 0.7719 | 0.7946 | 0.6422 | 0.0616 | 0.1631 | 0.5206 |
| | SVM | 0.7819 | 0.7781 | 0.6582 | 0.0460 | 0.1600 | 0.4974 |
| | XGBoost | 0.7824 | 0.8074 | 0.6760 | 0.0509 | 0.1540 | 0.4651 |
| S4 | TabResNet | **0.7698** | **0.8030** | **0.6469** | **0.0697** | **0.1655** | **0.5158** |
| | Ens_Hyper | **0.7808** | **0.8137** | **0.6629** | 0.0556 | **0.1592** | **0.5031** |
| | SVM | **0.7857** | **0.7973** | **0.6733** | **0.0424** | **0.1556** | **0.4878** |
| | XGBoost | **0.7900** | **0.8399** | **0.7065** | **0.0389** | **0.1471** | **0.4518** |

persona grouping (S2) causes the most significant degradation across all backbones. Excluding feature importance guidance in grouping (S1) or skipping the consistency check (S3) also weakens performance, though the impact is less severe. The full method (S4) consistently outperforms all ablated versions. For the curation ratio, the results in Figure 2(b) show a trend consistent with AIP: using the entire set of generated samples leads to inferior performance, whereas retaining about 30% of the high-confidence data achieves the most favorable trade-off across metrics.

## 4.4 PERFORMANCE UNDER OTHER LLMS

To assess the robustness of our framework under different backbone LLMs, we further evaluated LLM-OAP with *DeepSeek-V3*, *DeepSeek-R1*, and *Qwen3-plus* on AIP and BIP (see Tables 5 and 6). Across models, all three LLMs deliver consistent improvements over the non-augmented baselines,

with Qwen3-plus yielding overall balanced performance in both accuracy and predictive reliability. DeepSeek-V3 and DeepSeek-R1 also achieve competitive results, particularly in certain metrics such as AUC and AUCPR. These results suggest that the proposed group-based augmentation and curation scheme is largely agnostic to the choice of LLM, while the specific backbone may influence trade-offs across metrics. For the main paper, we report results with Qwen3-plus as it provides stable and strong performance across both datasets.

Table 5: Performance under different LLM models on AIP.

| LLM | Models | ACC ↑ | AUC ↑ | AUCPR ↑ | ECE ↓ | BS ↓ | NLL ↓ |
|---|---|---|---|---|---|---|---|
| DeepSeek-V3 | TabResNet | 0.8222 | 0.7697 | 0.5287 | 0.0487 | 0.1320 | 0.4253 |
| | Ens_Hyper | 0.8237 | 0.7637 | 0.5124 | 0.0389 | 0.1333 | 0.4280 |
| | SVM | 0.8338 | 0.7640 | 0.5243 | 0.0325 | 0.1304 | 0.4203 |
| | XGBoost | 0.8352 | 0.8020 | 0.5641 | 0.0332 | 0.1241 | 0.3982 |
| DeepSeek-R1 | TabResNet | 0.8300 | 0.7866 | 0.5489 | 0.0607 | 0.1285 | 0.4171 |
| | Ens_Hyper | 0.8280 | 0.7836 | 0.5458 | 0.0687 | 0.1298 | 0.4308 |
| | SVM | 0.8406 | 0.7765 | 0.5611 | 0.0309 | 0.1235 | 0.4041 |
| | XGBoost | 0.8348 | 0.8314 | 0.6085 | 0.0474 | 0.1174 | 0.3748 |
| QWEN3-plus | TabResNet | 0.8237 | 0.7765 | 0.5323 | 0.0475 | 0.1310 | 0.4245 |
| | Ens_Hyper | 0.8266 | 0.7889 | 0.5435 | 0.0408 | 0.1278 | 0.4105 |
| | SVM | 0.8372 | 0.7745 | 0.5436 | 0.0309 | 0.1256 | 0.4065 |
| | XGBoost | 0.8396 | 0.8198 | 0.5894 | 0.0319 | 0.1194 | 0.3834 |

Table 6: Performance under different LLM models on BIP.

| LLM | Models | ACC ↑ | AUC ↑ | AUCPR ↑ | ECE ↓ | BS ↓ | NLL ↓ |
|---|---|---|---|---|---|---|---|
| DeepSeek-V3 | TabResNet | 0.7692 | 0.7985 | 0.6515 | 0.0561 | 0.1626 | 0.5155 |
| | Ens_Hyper | 0.7717 | 0.8013 | 0.6467 | 0.0483 | 0.1601 | 0.4906 |
| | SVM | 0.7823 | 0.7871 | 0.6615 | 0.0427 | 0.1587 | 0.4907 |
| | XGBoost | 0.7827 | 0.8341 | 0.6926 | 0.0509 | 0.1494 | 0.4584 |
| DeepSeek-R1 | TabResNet | 0.7794 | 0.8209 | 0.6921 | 0.0747 | 0.1583 | 0.4975 |
| | Ens_Hyper | 0.7890 | 0.8085 | 0.6729 | 0.0783 | 0.1567 | 0.5071 |
| | SVM | 0.7953 | 0.8047 | 0.6929 | 0.0656 | 0.1520 | 0.4826 |
| | XGBoost | 0.7919 | 0.8495 | 0.7282 | 0.0640 | 0.1432 | 0.4450 |
| QWEN3-plus | TabResNet | 0.7698 | 0.8030 | 0.6469 | 0.0697 | 0.1655 | 0.5158 |
| | Ens_Hyper | 0.7808 | 0.8137 | 0.6629 | 0.0556 | 0.1592 | 0.5031 |
| | SVM | 0.7857 | 0.7973 | 0.6733 | 0.0424 | 0.1556 | 0.4878 |
| | XGBoost | 0.7900 | 0.8399 | 0.7065 | 0.0389 | 0.1471 | 0.4518 |

## 5 CONCLUSION

In this paper, we proposed LLM-OAP, an LLM-based data augmentation framework that integrates feature-aware persona grouping, behaviour-informed generation with confidence- and uncertainty-based curation for enhancing ML performance in order acceptance prediction. Experiments on real-world datasets with both full-information and limited-information settings show that our approach consistently outperforms GAN, diffusion, and LLM-based baselines, generally achieving the best or second-best results across all metrics. Ablation studies confirm the importance of each component, with moderate curation yielding the best trade-off between fidelity and diversity. In the future, we plan to enhance LLM-OAP with a reflection mechanism to further improve its performance and extend its applicability to broader predictive tasks.

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

# A   APPENDIX

## A.1   DETAILS OF ORIGINAL DATASET

The original dataset is derived from a stated preference (SP) survey designed to capture the ride acceptance behaviour of ride-sourcing drivers in both the United States and the Netherlands. The survey experiment was implemented under two information provision settings: *Baseline Information Provision (BIP)* and *Additional Information Provision (AIP)*. In the BIP scenario, only information currently available to drivers on existing platforms (e.g., Uber) is displayed, whereas in the AIP scenario, additional hypothetical information (e.g., guaranteed tip, traffic congestion, estimated fare) is revealed, allowing drivers to reassess the same request. This design enables comparison of behavioural responses to both existing and augmented information environments.

The features of the original dataset are summarized in Table 14, where we reordered the features such that environmental features precede driver-specific features. The table grouped them into decision variables, order attributes (BIP), additional attributes available in the AIP setting, driver-specific attributes and irrelevant features.

The BIP attributes replicate the current industry practice, such as the blind acceptance of trips without knowing fare or destination, while the AIP attributes introduce new monetary and contextual variables (e.g., guaranteed tip, traffic delay) to test their impact on drivers' acceptance probability.

## A.2   FEATURE IMPORTANCE

Table 7: Feature importance scores computed by permutation importance.

| Feature | Mean | Std | Feature | Mean | Std |
|---|---|---|---|---|---|
| Pickup | 0.0315 | 0.0079 | Partner | 0.0014 | 0.0014 |
| ID | 0.0168 | 0.0039 | Afternoon | 0.0010 | 0.0017 |
| Degree | 0.0048 | 0.0033 | Time1 | 0.0010 | 0.0018 |
| Morning | 0.0043 | 0.0020 | Experienced | 0.0010 | 0.0013 |
| EarnInc | 0.0043 | 0.0011 | Workhr | 0.0009 | 0.0029 |
| Acceptance | 0.0039 | 0.0023 | Cong | 0.0009 | 0.0032 |
| Beginners | 0.0033 | 0.0026 | NY | 0.0004 | 0.0007 |
| Fare | 0.0027 | 0.0039 | Midday | 0.0003 | 0.0017 |
| ExpInc | 0.0025 | 0.0030 | Full | 0.0001 | 0.0004 |
| Loc | 0.0025 | 0.0033 | Sat_Fri | 0.0000 | 0.0018 |
| Peak | 0.0022 | 0.0024 | Rate | 0.0000 | 0.0030 |
| Time2 | 0.0022 | 0.0026 | Night | 0.0000 | 0.0000 |
| Gender | 0.0019 | 0.0020 | NY_CA | -0.0003 | 0.0017 |
| Evening | 0.0014 | 0.0011 | Weekend | -0.0003 | 0.0017 |
| Peak_morning | 0.0014 | 0.0021 | Sat | -0.0006 | 0.0017 |
| Weekend_Friday | -0.0007 | 0.0019 | CA | -0.0007 | 0.0010 |
| Time | -0.0010 | 0.0035 | Peak_evening | -0.0012 | 0.0018 |
| Satisfied | -0.0013 | 0.0028 | Part | -0.0016 | 0.0025 |
| Long | -0.0020 | 0.0022 | Taxi | -0.0020 | 0.0032 |
| Age | -0.0020 | 0.0036 | Thu_Fri_Sat | -0.0020 | 0.0030 |
| Dec | -0.0022 | 0.0023 | Wait | -0.0032 | 0.0029 |
| Tip | -0.0043 | 0.0022 | Surge | -0.0045 | 0.0033 |
| Req | -0.0049 | 0.0013 | | | |

To identify the most influential features for order acceptance prediction, we trained a classifier model on the dataset and evaluated feature importance using permutation importance (Altmann et al., 2010).

For each feature $f$, its importance score was computed as the average decrease in model accuracy when the values of $f$ were randomly permuted while keeping other features fixed. Formally, let $Acc$ denote the baseline accuracy of the trained model on the test set, and $Acc_{\text{perm}}(f)$ denote the accuracy

after permuting feature $f$. The importance of feature $f$ is given by:

$$I(f) = Acc - \mathbb{E}[Acc_{\text{perm}}(f)] \tag{4}$$

where the expectation is estimated by averaging across multiple random permutations. The resulting scores were aggregated into a ranked table (see Table 7), from which the top 10 features were selected for further analysis.

To further examine whether the model fully utilizes the available feature space, we conducted an additional study by removing the bottom 15% of features ranked by permutation importance and re-evaluating XGBoost under both the AIP and BIP settings. The results are reported in Table 8.

Across both settings, removing these low-importance features consistently leads to performance degradation. For example, in the BIP scenario, AUCPR drops from 0.6623 to 0.6497 and AUC decreases from 0.8124 to 0.8022. Similarly, in the AIP setting, AUC decreases from 0.7940 to 0.7860 and AUCPR drops from 0.5471 to 0.5401. These results indicate that even features with relatively low measured importance still contribute to predictive performance, demonstrating that the model indeed leverages the full feature space rather than relying solely on a few dominant features.

Table 8: Effect of removing the last 15% lowest-importance features on XGBoost performance (AIP and BIP settings).

| Model | ACC ↑ | AUC ↑ | AUCPR ↑ | ECE ↓ | BS ↓ | NLL ↓ |
|---|---|---|---|---|---|---|
| XGBoost (BIP) | 0.7596 | 0.8124 | 0.6623 | 0.0519 | 0.1569 | 0.4781 |
| XGBoost w/o last 15% features (BIP) | 0.7477 | 0.8022 | 0.6497 | 0.0579 | 0.1595 | 0.4821 |
| XGBoost (AIP) | 0.8237 | 0.7940 | 0.5471 | 0.0457 | 0.1270 | 0.4023 |
| XGBoost w/o last 15% features (AIP) | 0.8161 | 0.7860 | 0.5401 | 0.0478 | 0.1277 | 0.4068 |

### A.3 CONFIDENCE AND UNCERTAINTY DISTRIBUTION OF SYNTHETIC DATA

Figure 3 illustrates the confidence and uncertainty distributions of the LLM-generated synthetic dataset both before and after applying our curation scheme. Before curation, the synthetic samples exhibit clear irregularities: the confidence distribution is highly bimodal, with a large fraction of samples concentrated near the extremes (very low or very high confidence). Likewise, the corresponding uncertainty values skew toward the upper range. These patterns indicate that many raw synthetic samples are either poorly aligned with the underlying decision boundary or exhibit unstable predictions, suggesting a high level of noise and unreliability if used directly for model training.

After applying our confidence- and uncertainty-based curation filter, the distributions shift markedly. The retained samples show greater concentration in the high-confidence, low-uncertainty region, while extreme low-quality samples are effectively removed. This demonstrates that the curation step successfully suppresses noisy or ambiguous synthetic instances and yields a more reliable augmented dataset with improved fidelity. The contrast between the before- and after-curation distributions empirically validates the necessity and effectiveness of our curation strategy in stabilizing the quality of LLM-generated tabular data.

### A.4 PROMPT DESIGN

Figure 6 illustrates an example of the prompts used in LLM-OAP. The prompt is structured in three stages: (1) behaviour summary, where the LLM is instructed to summerize the order acceptance tendencies of a given persona group based on its attributes, acceptance/reject rate, and representative records; (2) order sythesis, where the LLM generates a set of realistic ride-hailing orders conditioned on the same persona attributes and examples, along with the allowable driver IDs and their associated attributes; and (3) decision simulation, where the LLM is provided with the preceding dialogue history and tasked with assigning decision labels (accept/reject) to the sythesized orders according to the summerized behavioural patterns. This staged design ensures that the generated data remain logically consistent and aligned with observed driver preferences.

To clarify the variables used in the prompt template, the placeholders correspond to group-specific or sample-specific quantities extracted from the dataset. Specifically, {`persona_attr`} denotes

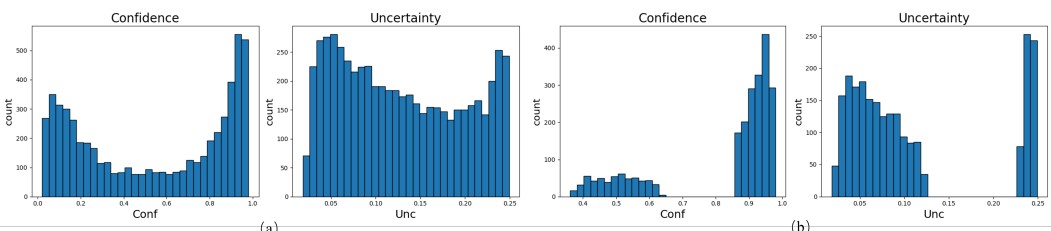

Figure 3: Distribution of synthetic samples generated by LLM before(a) and after(b) curation.

the categorical attributes that define the persona group (e.g., Age_group, Beginners, NY_CA, Part); {n_records} is the number of real historical samples contained in that group; {rk} represents the empirical acceptance rate; {examples} contains several representative records sampled from the group; {id_attr} lists allowable driver IDs together with their associated attributes; {n} specifies the number of synthetic orders to be generated; and {orders} denotes the sequence of synthesised orders that require decision labeling. These variables ensure that each prompt is grounded in real data and that the LLM generation remains persona-consistent and behaviourally coherent.

## A.5 STABILITY ANALYSIS AND STATISTICAL SIGNIFICANCE TESTING

To assess the robustness of our method LLM-OAP, we conducted two complementary analyses: (1) variability across repeated LLM generation runs, and (2) statistical significance testing comparing models trained with and without synthetic data.

**(1) Variance across repeated LLM generations.** We prompted the LLM to generate the synthetic dataset three independent times under identical configurations. For each dataset, we trained an XGBoost classifier and evaluated it on the same held-out test set. We then computed the mean, variance, standard deviation, and 95% confidence intervals for six evaluation metrics (ACC, AUC, AUCPR, ECE, BS, and NLL). As shown in Table 9, all metrics exhibit extremely low variance and narrow confidence intervals, indicating that the downstream model performance is highly stable and robust to randomness in the LLM generation process.

**(2) Statistical significance of performance improvements.** Beyond stability analysis, we evaluated whether the performance gains brought by synthetic data are statistically meaningful. We compared an XGBoost model trained solely on the original dataset with one trained on both original and synthetic samples. Using the same test set, we applied: (i) 2,000-sample *stratified bootstrap tests* for AUCPR, AUC, and ECE (percentile 95% CI, two-sided $p$-values), (ii) a *paired t-test* and a *Wilcoxon signed-rank test* for NLL, and (iii) *McNemar's test* for Accuracy. The results in Table 10 reveal statistically significant improvements in AUCPR (diff = 0.0798, $p < 0.001$), AUC (diff = 0.0441, $p < 0.001$), NLL ($p = 0.0023$; Wilcoxon $p = 2.3 \times 10^{-7}$), and ACC ($p = 0.021$). ECE shows a mild improvement trend but is not statistically significant. These results confirm that the benefits of synthetic data are both stable across generations and statistically significant.

## A.6 VALIDATION ON REAL REVEALED-PREFERENCE (RP) DATA

To examine the external validity and real-world applicability of our framework, we additionally evaluated LLM-OAP on a real revealed-preference (RP) delivery dataset. Unlike SP data, RP data reflect couriers(riders)' actual historical acceptance decisions collected from real delivery logs, thus serving as a strong test of practical behavioural fidelity.

Table 9: Performance variability across three independent synthetic data generation runs.

| Metric | Mean | Variance | Std | 95% CI |
|--------|------|----------|-----|--------|
| ACC | 0.8420 | 0.000005 | 0.00219 | [0.83655, 0.84745] |
| AUC | 0.8223 | 0.000125 | 0.01116 | [0.79457, 0.85003] |
| AUCPR | 0.5901 | 0.000006 | 0.00247 | [0.58394, 0.59620] |
| ECE | 0.0331 | 0.000001 | 0.00111 | [0.03032, 0.03581] |
| BS | 0.1200 | 0.000005 | 0.00222 | [0.11452, 0.12555] |
| NLL | 0.3848 | 0.000052 | 0.00720 | [0.36691, 0.40269] |

Table 10: Statistical significance tests comparing performance gains with synthetic data.

| Metric | Difference (Ours – Baseline) | Statistical Test Result |
|--------|------------------------------|--------------------------|
| ACC | – | $p = 0.0212$ (McNemar) |
| AUC | 0.0441 | 95% CI: [0.0191, 0.0712],   $p < 0.001$ (bootstrap) |
| AUCPR | 0.0798 | 95% CI: [0.0296, 0.1308],   $p < 0.001$ (bootstrap) |
| ECE | –0.0107 | 95% CI: [–0.0323, 0.0118],   $p = 0.35$ (bootstrap) |
| NLL | –0.0394 | $p = 0.0023$ (paired t-test),   Wilcoxon $p = 2.3 \times 10^{-7}$ |

The RP dataset is obtained from the publicly available *INFORMS TSL Data Challenge* repository[1]
, which provides real-world delivery order assignment data and courier acceptance behaviour, contributed by Meituan for research on operational decision-making.

We trained an XGBoost classifier on the original RP dataset and compared it with an XGBoost model trained on RP data augmented using our LLM-OAP framework. The results, summarized in Table 11, show substantial performance improvements across all major metrics: ACC, AUC, AUCPR, BS, and NLL. In particular, AUCPR increases from 0.9808 to 0.9989, and NLL decreases from 0.2449 to 0.1573, demonstrating that LLM-OAP remains effective when applied to real-world behavioural data.

These findings confirm that our method generalizes beyond SP settings and can provide meaningful benefits in real operational environments.

Table 11: Performance on real RP data.

| Model | ACC ↑ | AUC ↑ | AUCPR ↑ | ECE ↓ | BS ↓ | NLL ↓ |
|-------|-------|-------|---------|-------|------|-------|
| XGBoost | 0.9022 | 0.8966 | 0.9808 | 0.0275 | 0.0727 | 0.2449 |
| **XGBoost w/ LLM-OAP** | **0.9657** | **0.9927** | **0.9989** | **0.0448** | **0.0404** | **0.1573** |

## A.7 SHAP-BASED INTERPRETABILITY ANALYSIS

To improve the interpretability, we apply SHAP (SHapley Additive exPlanations) to the trained XGBoost model. Using TreeExplainer, we compute SHAP values on the test set and visualize them through a summary bar plot and a summary dot plot, which respectively show the average magnitude of each feature's contribution and the distribution of its directional effects.

As shown in Figure 4, ID and Degree emerge as the most influential variables, indicating that latent individual heterogeneity and socioeconomic status strongly shape acceptance decisions. Income-related attributes such as EarnInc and ExpInc also exhibit substantial impact, while contextual and temporal factors—including Weekend_Friday, Age, and Workhr—meaningfully contribute to the model's output. Incentive-related features (Tip, Surge, Fare) generally increase acceptance probability, whereas effort- or cost-related factors such as Wait, Cong, and Long decrease it. The color-coded dot plot further reveals substantial heterogeneity across samples: high feature values often

---

[1] https://connect.informs.org/tsl/tslresources/datachallenge

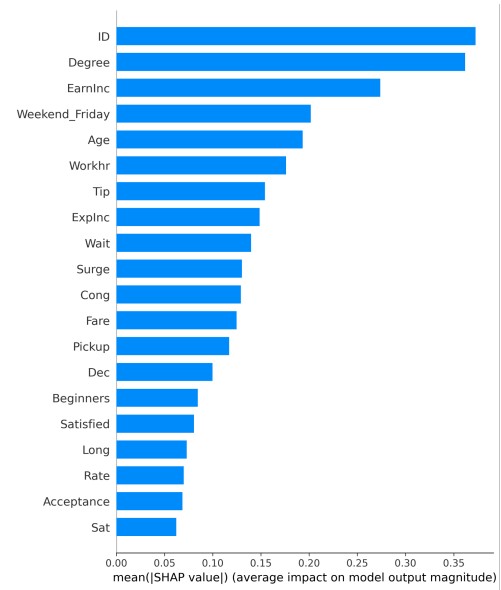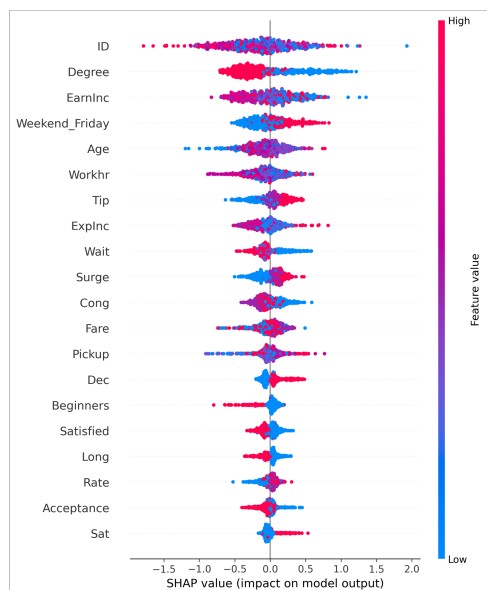

Figure 4: SHAP summary plots showing global feature importance (left) and the distribution of feature effects (right) for the XGBoost classifier.

shift predictions in intuitive directions, yet the broad dispersion of SHAP values across drivers reflects diverse behavioral responses to identical conditions. Together, the two SHAP visualizations demonstrate that the model captures a coherent and interpretable structure of decision drivers while retaining nuanced individual-level variability.

With both feature permutation importance and SHAP analysis, we summarize that the model consistently prioritizes socio-economic factors, driver-specific heterogeneity, and key contextual variables, while low-importance features contribute only marginal effects, confirming that the predictor learns a stable and interpretable decision structure rather than relying on spurious correlations.

## A.8 EFFECT OF RETAINING HIGH-UNCERTAINTY SAMPLES

To examine the role of high-uncertainty samples in the curation stage, we conduct an ablation study by varying the proportion of retained uncertain samples while fixing the high-confidence retention ratio at 0.3. As shown in Figure 5, increasing the uncertainty ratio from 0 to 0.10 consistently improves overall predictive performance: accuracy (ACC), AUC, and AUCPR all rise to their peak values at an uncertainty ratio of 0.10, indicating that a small amount of uncertain but informative samples can enhance model generalization. Meanwhile, calibration-related metrics (ECE, BS, NLL) monotonically decrease up to the 0.10 setting, suggesting improved probability calibration and reduced predictive uncertainty. When the uncertainty ratio further increases to 0.20, both performance and calibration metrics slightly degrade, implying that excessive uncertain samples introduce noise rather than useful diversity. Overall, these results show that a moderate inclusion of high-uncertainty samples (around 0.10) is beneficial, whereas retaining too many uncertain instances weakens the robustness and calibration of the model.

## A.9 SENSITIVITY ANALYSIS ON FEATURE GROUPING STRATEGY

To determine the most suitable feature combination for persona grouping, we conducted a series of experiments across six candidate feature sets (F1–F6). These sets were designed using two strategies: (1) selecting features that are both domain-relevant and ranked highly in permutation importance, and (2) constructing random combinations to avoid bias from manual feature selection.

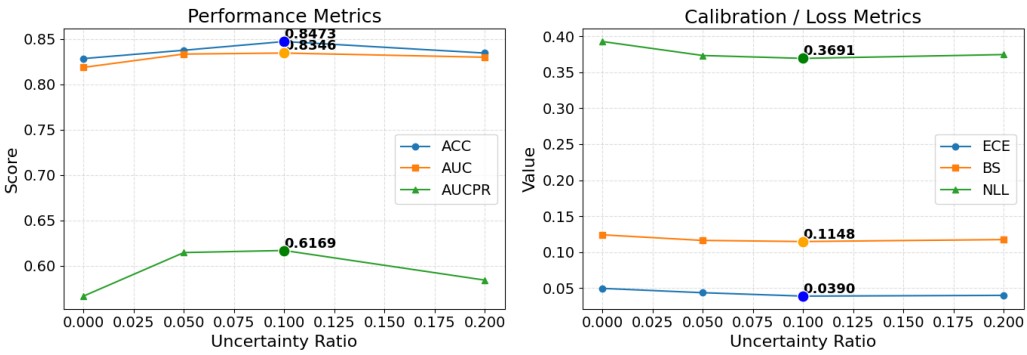

Figure 5: Effect of Retaining High-Uncertainty Samples.

Our goal was to identify a feature set that can meaningfully distinguish heterogeneous behavioral patterns while avoiding excessive within-group imbalance.

Table 12 summarizes the results of these experiments. For each feature set, we report the total number of groups formed, the variance of the group-wise acceptance rates $\text{Var}(p_{\text{accept}})$, the number of groups that are fully imbalanced (i.e., containing only acceptances or only rejections), and a normalized imbalance score that measures overall within-group imbalance. A larger number of groups indicates finer behavioral segmentation, but it also leads to smaller group sizes and increases the likelihood of extreme imbalance. Conversely, when too few features are used for grouping, the resulting groups become overly coarse and fail to capture the diverse behavioral patterns present in the dataset.

The results reveal a clear trade-off. Feature sets with many attributes (e.g., F1 with six features) produce very fine-grained partitioning, but at the cost of numerous highly imbalanced or extremely small groups, which would provide unreliable or misleading behavioral signals to the LLM during conditional generation. On the other hand, minimal feature sets (e.g., F5 with only two features) drastically reduce imbalance but collapse many distinct driver patterns into overly broad groups, undermining the purpose of persona-aware generation.

A middle-sized feature subset offers the best balance between expressiveness and stability. Among all evaluated combinations, F6 (Age_group, Beginners, NY_CA, Part) stands out as the optimal choice: it produces 32 groups—enough to capture meaningful heterogeneity—while maintaining only one fully imbalanced group and exhibiting low acceptance-rate variance and imbalance scores. This configuration preserves salient behavioral differences while ensuring each group remains representative and well-balanced for conditional prompting.

Based on these findings, we adopt F6 as the feature set for persona grouping, as it achieves the desired balance between behavioral distinctiveness and group stability, thereby providing the most reliable structure for downstream LLM-conditioned data generation.

Table 12: Summary of persona grouping experiments across six feature sets.

| Feature Set | #Groups | $\text{Var}(p_{\text{accept}})$ | #Imbalanced Groups | Imbalance Score |
|---|---|---|---|---|
| F1 | 117 | 0.0304 | 18 | 1.0000 |
| F2 | 63 | 0.0281 | 5 | 0.9661 |
| F3 | 33 | 0.0273 | 5 | 0.7419 |
| F4 | 18 | 0.0226 | 2 | 0.5396 |
| F5 | 10 | 0.0248 | 0 | 0.5689 |
| F6 | 32 | 0.0194 | 1 | 0.5417 |

**Feature sets:** F1 = {Age_group, Degree, Acceptance, Peak, Gender, Morning}; F2 = {Age_group, Degree, Acceptance, Peak, Gender}; F3 = {Age_group, Acceptance, Beginners, Degree}; F4 = {Age_group, Acceptance, Beginners}; F5 = {Age_group, Acceptance}; F6 = {Age_group, Beginners, NY_CA, Part}.

## A.10 COMPARISON WITH SMOTENC

To further validate that the performance gains of LLM-OAP do not simply arise from oversampling, we conduct an additional comparison using SMOTENC, a widely used extension of SMOTE designed for mixed numerical–categorical tabular data. SMOTENC synthesizes new samples by interpolating numerical attributes while sampling categorical attributes based on nearest neighbors.

We trained an XGBoost classifier using SMOTENC-augmented data under both AIP and BIP scenarios and compared its performance with our LLM-OAP. The results are shown in Table 13. Unlike LLM-OAP—which explicitly models feature interactions and produces structured behavioral patterns—SMOTENC does not improve predictive performance substantially across any of the six evaluation metrics. In contrast, LLM-OAP yields consistently higher AUC and AUCPR and substantially lower ECE, BS, and NLL.

These results confirm that the benefits of LLM-OAP are not attributable to simple resampling. Instead, improvements stem from the LLM's ability to generate behaviourally coherent synthetic samples that enrich the feature space in ways traditional tabular oversampling methods cannot.

Table 13: Comparison between SMOTENC and LLM-OAP on XGBoost.

| Scenario | Method | ACC ↑ | AUC ↑ | AUCPR ↑ | ECE ↓ | BS ↓ | NLL ↓ |
|----------|--------|-------|-------|---------|-------|------|-------|
| AIP | SMOTENC | 0.8133 | 0.7908 | 0.5219 | 0.0903 | 0.1314 | 0.4227 |
| AIP | LLM-OAP | **0.8396** | **0.8198** | **0.5894** | **0.0319** | **0.1194** | **0.3834** |
| BIP | SMOTENC | 0.7621 | 0.8109 | 0.6598 | 0.1023 | 0.1561 | 0.4737 |
| BIP | LLM-OAP | **0.7900** | **0.8399** | **0.7065** | **0.0389** | **0.1471** | **0.4518** |

## A.11 USE OF LLM IN MANUSCRIPT PREPARATION

We acknowledge that an LLM was employed to assist in polishing the writing of this manuscript. The LLM was used exclusively for language refinement (e.g., grammar, clarity, and style).

Below are the attributes and real samples of a given Persona group:
Persona attributes: {persona_attr}
Numbers of examples: {n_records}
Acceptance rate: {rk}
Rejection rate: {1 - rk}
Example records: {examples}
Please summarize the order acceptance behaviour patterns of the group.

e.g. Drivers in this Persona group are aged 20-29, not beginners, not located in New York or California, and not working part-time. Their overall acceptance rate is moderate to low, with a relatively high rejection rate. They tend to accept orders in the evening and nighttime, respond more positively to peak-hour requests, and prefer short-distance trips with low waiting times, minimal congestion, or lower tips. In contrast, they are more likely to reject long-distance orders, those with long waiting times, or those involving traffic congestion.

Below are the attributes and real samples of a given Persona group:
Persona attributes: {persona_attr}
Numbers of examples: {n_records}
Allowable driver IDs and their associated attributes: {id_attr}
Acceptance rate: {rk}
Rejection rate: {1 - rk}
Example records: {examples}
Please generate {n} reasonable, realistic ride-hailing orders without decision labels based on the above information.

e.g. [{"Req": 1, "Time": 1, "Time1": 1, "Time2": 0, "Wait": 10, "Dec": 1,...}, ...]

Please add decision labels to the following orders based on the previous conversations:
Orders: {orders}

e.g. [{"Choice1":1, "Choice2": 1, "Req": 1, "Time": 1, "Time1": 1, "Time2": 0, "Wait": 10, "Dec": 1,...}, ...]

Figure 6: A prompt example of our method LLM-OAP.

Table 14: Feature descriptions of the original dataset.

| Field | Description |
|---|---|
| *Decision Variables* | |
| Choice1 | Order acceptance in BIP scenario (2=accept, 1=reject) |
| Choice2 | Order acceptance in AIP scenario (2=accept, 1=reject) |
| *Order Attributes (BIP)* | |
| Req | Request type (0=Uber X, 1=Uber Pool) |
| Time | Period within shift (1=start, 2=middle, 3=end) |
| Time1 | Whether it is the beginning of a shift (1=yes, 0=no) |
| Time2 | Shift length (0=4h, 1=8h) |
| Wait | Waiting time between orders (minutes: 0/5/15) |
| Dec | Previous order rejected (1=yes, 0=no) |
| Rate | Average passenger rating of driver (3/4/5) |
| Pickup | Travel time to passenger pickup location (minutes: 5/10/15/20) |
| Loc | Driver location (0=suburb, 1=city center) |
| Surge | Surge bonus (0/1.5/3) |
| Long | Whether trip duration exceeds 30 minutes (1=yes, 0=no) |
| *Additional Attributes (AIP only)* | |
| Cong | Estimated traffic delay (minutes: 0/15/30) |
| Tip | Guaranteed tip (0/1.5/3) |
| Fare | Estimated fare (8/16/24) |
| *Driver Attributes* | |
| Acceptance | Above historical average acceptance rate (1=yes, 0=no) |
| Workhr | Working hours |
| Part | Part-time driver (1=yes, 0=no) |
| Full | Full-time driver (1=yes, 0=no) |
| Age | Driver's age |
| ID | Driver identifier |
| Beginners | Driver with less than 12 months of experience (1=yes, 0=no) |
| Experienced | Experienced driver (1=yes, 0=no) |
| Satisfied | Fully satisfied with platform (rating $\geq$4.5/5, 1=yes, 0=no) |
| Taxi | Taxi driving experience (1=yes, 0=no) |
| Gender | Gender (1=male, 0=female) |
| Partner | Marital/partner status (1=partner, 0=single) |
| Degree | Education (1=college or above, 0=no) |
| NY | Located in New York (1=yes, 0=no) |
| CA | Located in California (1=yes, 0=no) |
| NY_CA | Located in NY or CA (1=yes, 0=no) |
| EarnInc | Perceived income change during pandemic |
| ExpInc | Perceived workload/order change during pandemic |
| Morning | Shift starts in morning (5–11h) (1=yes, 0=no) |
| Midday | Shift starts at midday (11–15h) (1=yes, 0=no) |
| Afternoon | Shift starts in afternoon (15–19h) (1=yes, 0=no) |
| Evening | Shift starts in evening (19–23h) (1=yes, 0=no) |
| Night | Shift starts at night (23–5h) (1=yes, 0=no) |
| Weekend | Typical working day is weekend (1=yes, 0=no) |
| Weekend_Friday | Typical working day is weekend or Friday (1=yes, 0=no) |
| Sat_Fri | Typical working day is Saturday or Friday (1=yes, 0=no) |
| Sat | Typical working day is Saturday (1=yes, 0=no) |
| Thu_Fri_Sat | Typical working day is Thursday, Friday, or Saturday (1=yes, 0=no) |
| Peak_morning | Working during morning peak hours (1=yes, 0=no) |
| Peak_evening | Working during evening peak hours (1=yes, 0=no) |
| Peak | Working during peak hours (1=yes, 0=no) |
| *Irrelevant Features (Removed)* | |
| Block | Survey block ID |
| Fac1000 | Sliding-window factor (1000-unit fatigue/system feature) |
| Fac2000 | Sliding-window factor (2000-unit fatigue/system feature) |

