# OpenReview forum: "LLM-OAP: LLM-based Data Augmentation Framework for Enhancing Order Acceptance Prediction in Mobility-on-Demand Systems"
_ICLR.cc/2026/Conference — Submitted to ICLR 2026_

### Official Review · Reviewer_s4dK · 2025-10-25

**Soundness:** 2
**Presentation:** 2
**Contribution:** 2
**Rating:** 4
**Confidence:** 3

**Summary:**

This paper presents LLM-OAP, a data-augmentation framework that leverages Large Language Models (LLM) to produce high-quality synthetic samples for small-scale behavioural datasets, with the goal of improving order-acceptance prediction in Mobility-on-Demand (MoD) systems. Drivers are first clustered into personas via feature-aware grouping; synthetic orders and corresponding behavioural labels are then generated, after which a confidence-based and uncertainty-based filter retains only the most reliable samples. Evaluations on stated-preference (SP) survey data, under both full-information and limited-information regimes, show consistent gains over strong ML baselines, GAN / diffusion augmentations and recent LLM-centric methods. Extensive ablations and empirical analyses are offered to support interpretability and generalisability claims.

**Strengths:**

• Tackles a commercially relevant, technically hard problem: scarce data, distribution gaps and non-linear driver behaviour in MoD platforms.
• Proposes a novel, fully structured LLM pipeline that mitigates data scarcity, strong subjectivity and the limited external validity typical of SP surveys.
• Combines persona-based grouping, explicit behavioural summarisation, synthetic-order generation and label simulation, capturing heterogeneous driver preferences instead of forcing a single model on all users. Figure 1 convincingly illustrates the design.
• Confidence / uncertainty filtering demonstrably reduces noisy or biased samples; Figure 3 quantifies the denoising effect.
• Experiments are meticulous: Tables 1 & 3 report accuracy and calibration metrics (ACC, AUC, AUCPR, ECE, BS, NLL) under both information regimes, beating GAN, diffusion and LLM baselines.
• Tests multiple LLM back-ends (DeepSeek-V3, R1, Qwen3-plus); performance is largely invariant to the choice, indicating robustness.
• Comprehensive baselines (GAN, diffusion, other LLM schemes) and ablations (Tables 2 & 4) together with filtering-ratio analysis (Figure 2) isolate the contribution of each module.
• Figures are polished and informative; prompt designs are disclosed in Figure 4, enhancing reproducibility.
• Writing is clear; related work is well positioned.

**Weaknesses:**

1. External validity: all experiments use SP data; no validation on large-scale real order logs. The framework’s practical impact on revealed-preference (RP) environments remains unverified.
2. Potential SP over-fitting: the XGBoost filter is trained only on the original SP set, risking preservation of SP-specific artefacts rather than true behavioural realism.
3. Diversity evaluation is missing: no quantitative metric or visualisation of behavioural diversity post-augmentation; aggressive filtering may inadvertently over-represent dominant patterns.
4. Feature grouping lacks justification: random selection from the top-10 features (§3.2) is ad-hoc; no sensitivity analysis on group size or feature choice.
5. Black-box LLM decisions: no theoretical or empirical evidence that synthetic labels possess behavioural fidelity; consistency checks against human drivers or RP data are absent.
6. LLM hallucination and adversarial robustness are not examined. And no explicit stress tests or human audits of harmful or contradictory samples.
7. Reproducibility gaps: hyper-parameters of the filtering stage, pre- filtering statistics, post-filtering statistics and exact prompt variants are only partially reported.
8. Small data scale: the total augmented volume is tiny compared with industrial MoD data; many features show negligible permutation importance, suggesting under-utilisation of the feature space.
9. Unequal post-generation filtering: GAN and diffusion baselines are not subjected to the same confidence-based pruning, confounding the comparison.
10. Limited interpretability: despite claims, no model-level explanations (e.g., SHAP, attention heat-maps) are supplied for either the final predictor or the generation process.
11. Figures could be richer: Fig. 2 omits diversity curves; Fig. 3 shows pre- filter but not post-filter distributions.
12. Cost concerns: large-scale LLM API calls for data generation and label simulation may be prohibitively expensive for production deployment.

**Questions:**

See weaknesses.

---

> ### Author Response · Authors · 2025-11-26
> **Response to Reviewer s4dK (1/6)**
>
> > W1. External validity: all experiments use SP data; no validation on large-scale real order logs. The framework’s practical impact on revealed-preference (RP) environments remains unverified.
>
> A: Thank you for raising this point. We agree that external validity and performance on revealed-preference (RP) data are important. Our approach, however, is not limited to SP data, it can also be applied to RP data.
>
> In the original submission, we used SP data to predict driver order acceptance behaviour, because driver level order acceptance logs from ride-hailing platforms are typically owned and controlled by companies like Uber, Didi, which are subject to strict privacy constraints, and difficult to share for reproducible research. SP data, in contrast, provide clean, fully observed accept/reject decisions with controlled attribute variation, which makes it a natural and reproducible starting point. LLM-OAP is designed to expand this limited but high-quality SP dataset into a larger, behaviourally coherent training set, so that existing ML models (e.g., XGBoost, TabResNet) can realise their potential in order-acceptance prediction.
>
> While the main experiments use SP data, our framework itself is not limited to SP settings; it operates by extracting and reconstructing behavioural structures, which is equally applicable to real-world RP environments. To assess external validity, we conducted additional experiments on real RP order logs and included the results in the appendix A.6. The RP dataset is obtained from the publicly available INFORMS TSL Data Challenge repository, which contains real-world delivery order assignment and courier(rider) decision records provided by Meituan. Given the strong performance of XGBoost, we apply our framework to enhance it for tackling the RP environment. For your convenience, we present the results below for your reference:
>
> | Model                 | ACC ↑    | AUC ↑    | AUCPR ↑  | ECE ↓    | BS ↓    | NLL ↓   |
> |-----------------------|----------|----------|----------|----------|---------|---------|
> | XGBoost               | 0.9022   | 0.8966   | 0.9808   | 0.0275   | 0.0727  | 0.2449  |
> | **XGBoost w/ LLM-OAP** | **0.9657** | **0.9927** | **0.9989** | **0.0448** | **0.0404** | **0.1573** |
>
> According to the results, we observed that XGBoost performance improves from 0.9022 / 0.8966 / 0.9808 to 0.9657 / 0.9927 / 0.9989 (ACC / AUC / AUCPR), demonstrating that the framework transfers effectively to revealed-preference scenarios. This indicates that our method has practical value for real operational systems beyond the specific SP dataset used in the main experiments.

---

> ### Author Response · Authors · 2025-11-26
> **Response to Reviewer s4dK (2/6)**
>
> > W2. Potential SP over-fitting: the XGBoost filter is trained only on the original SP set, risking preservation of SP-specific artefacts rather than true behavioural realism.
>
> A: Thanks for your comment. First, we clarify that the SP dataset used in our paper is collected from real human surveys [1], and therefore already reflects genuine behavioural tendencies rather than synthetic or simulated patterns.
>
> Second, the goal of our curation mechanism (i.e., XGBoost filter) is **not** to learn SP-specific artefacts, but to filter out low-quality LLM-generated samples using model-estimated confidence and uncertainty. We retain high-confidence, low-uncertainty samples and intentionally preserve a small portion (~10%) of high-uncertainty samples to maintain diversity, so that we can improve the final prediction performance. Our ablation study (Figure 2 and Appendix A.8) has shown that removing this mechanism or altering the retention ratios leads to clear performance degradation, confirming that our curation mechanism with its proper configurations improves data quality for boosting prediction performance, rather than reinforcing dataset-specific artefacts.
>
> Third, we have developed effective components to preserve behavioural realism **throughout the generation stage**: 1) Persona-conditioned prompting encourages the LLM to model group-specific behavioural patterns, which facilitates fine-grained and realistic behaviour generation. 2) Our driver-attribute consistency check eliminates hallucinated samples, in which the demographic attributes contradict those of the corresponding driver. This scheme ensures that the generated samples follow the real survey data. These components together ensure that the LLM-generated samples remain realistic and coherent with the original data. The ablation studies in section 4 (Table 2 on page 8 and Table 4 on page 9) have shown the effectiveness of these components.
>
> Last but not least, our results (Table 1 on page 7 and Table 3 on page 9) show that training models with these enhanced samples leads to significant performance gains on the test set (**Note: the test set consists entirely of realistic data collected from the real human survey**.). It implies that our framework with the components results in true behavioural realism, which enhances the performance of predicting the real human behaviours.
>
> [1] Ashkrof P, de Almeida Correia G H, Cats O, et al. Ride acceptance behaviour of ride-sourcing drivers[J]. Transportation Research Part C: Emerging Technologies, 2022, 142: 103783.
>
> > W3. Diversity evaluation is missing: no quantitative metric or visualisation of behavioural diversity post-augmentation; aggressive filtering may inadvertently over-represent dominant patterns.
>
> A: Thank you for pointing this out. During curation (i.e., filtering) stage, we did **not** adopt an aggressive strategy that keeps only high-confidence samples. Instead, to ensure the behavioural diversity, we intentionally retained some high-uncertainty samples.
>
> Our ablation study in appendix A.8 (on page 18) has shown that by applying the uncertain samples in the confidence-based filtering, we can consistently improve downstream performance by retaining the behavioural diversity post-augmentation. Specifically, a moderate inclusion of such uncertain samples improves overall performance, whereas keeping none or too many leads to degradation. These results confirm that our uncertainty-based curation scheme maintains diversity while ensuring high-quality augmentation.
>
> Besides the significant performance improvements, following your suggestion, we now explicitly quantify drivers’ behavioural diversity. In specific, we calculate the entropy of the drivers’ behaviours in the original dataset and that of the augmented dataset. We observed that the entropy after augmentation remains close to that of the original dataset (0.68 vs. 0.75), indicating that the behavioural diversity is largely preserved. We acknowledge that more specialised metrics could be used to further assess the generative diversity of LLM-based augmentation; a systematic comparison is beyond the scope of this work.
>
> In summary, all our results with analysis demonstrate that our filtering achieves a good trade-off between the data diversity and the performance improves across all ML and DL models.

---

> ### Author Response · Authors · 2025-11-26
> **Response to Reviewer s4dK (3/6)**
>
> > W4. Feature grouping lacks justification: random selection from the top-10 features (§3.2) is ad-hoc; no sensitivity analysis on group size or feature choice.
>
> A: Thanks for your suggestion. In our original submission, we select the group size and features according to exploratory results. **Now we have added a more detailed sensitivity analysis in Appendix A.9 (on page 18) to justify our feature-grouping design**. Specifically, we evaluated six different feature sets (which are selected by both importance-guided and randomly constructed combinations) and compared their effects on the number of groups, within-group balance, and behavioural separability. For your convenience, we present the results below for your reference. As shown, there exists a clear trade-off: using too many features leads to more groups, which are fragmented and highly imbalanced, while using too few collapses distinct behavioural patterns with little variance of human behaviours (accept or reject).
>
> Based on the analysis and the results, we selected the feature set (Age_group, Beginners, NY_CA, Part) because it provides the best balance among all feature sets, thereby capturing heterogeneous behaviour and maintaining well-balanced groups. According to our results, this feature set delivers strong performance with augmented dataset, comprehensively enhancing all models in predicting human behaviours.
>
> ### Summary of persona grouping experiments across six feature sets
>
> | Feature Set | #Groups | Variance(p_accept) | #Imbalanced Groups | Imbalance Score |
> |-------------|---------|----------------|----------------------|------------------|
> | F1 | 117 | 0.0304 | 18 | 1.0000 |
> | F2 | 63  | 0.0281 | 5  | 0.9661 |
> | F3 | 33  | 0.0273 | 5  | 0.7419 |
> | F4 | 18  | 0.0226 | 2  | 0.5396 |
> | F5 | 10  | 0.0248 | 0  | 0.5689 |
> | F6 | 32  | 0.0194 | 1  | 0.5417 |
>
> **Feature sets:**
> - **F1** = {Age_group, Degree, Acceptance, Peak, Gender, Morning}
> - **F2** = {Age_group, Degree, Acceptance, Peak, Gender}
> - **F3** = {Age_group, Acceptance, Beginners, Degree}
> - **F4** = {Age_group, Acceptance, Beginners}
> - **F5** = {Age_group, Acceptance}
> - **F6** = {Age_group, Beginners, NY_CA, Part}
>
> > W5. Black-box LLM decisions: no theoretical or empirical evidence that synthetic labels possess behavioural fidelity; consistency checks against human drivers or RP data are absent.
>
> A: Thank you for raising this question. First, we clarify that all ground-truth labels in our evaluation are real human decisions. The SP dataset consists of acceptance/rejection choices made by real drivers in a stated-preference survey, and the additional RP experiment (Appendix A.6) uses real driver decisions from the INFORMS TSL delivery dataset. Synthetic labels generated by LLM-OAP are used only for training, and all reported metrics are computed on a separate test set containing only real human-labelled data. Our synthetic data are not produced purely through unconstrained black-box LLM generation. Instead, we enforce behavioural structure and apply a consistency check that removes any hallucinated or behaviourally implausible samples. Specifically, we verify that every LLM-generated record remains fully consistent with the human driver’s original personal attributes (e.g., experience level, location, working status). This prevents the LLM from inventing contradictory or impossible attribute-behaviour combinations, thereby ensuring that generated choices remain faithful to the driver’s true profile. The Ablation Study S3 (on page 7 and 8) shows that removing this check leads to clear performance degradation (Tables 2 and 4), demonstrating its necessity. Therefore, our framework explicitly promotes behavioural fidelity and provides empirical evidence that the generated labels are consistent with human driver characteristics.

---

> ### Author Response · Authors · 2025-11-26
> **Response to Reviewer s4dK (4/6)**
>
> > W6. LLM hallucination and adversarial robustness are not examined. And no explicit stress tests or human audits of harmful or contradictory samples.
>
> A: Thanks for pointing this out. Even though we do not run dedicated adversarial stress tests, our framework already incorporates several mechanisms that directly mitigate LLM hallucination and improve robustness.
>
> First, as described in section 3.2 (on page 5), the **ID-attribute consistency check** ensures that every generated sample strictly matches the driver’s original immutable attributes (e.g., experience level, part-time status, region), preventing the LLM from producing contradictory or behaviourally impossible profiles.
>
> Second, the **confidence- and uncertainty-based curation** systematically filters out low-quality or unstable generations, which implicitly removes hallucinated or logically inconsistent outputs.
>
> Third, the **group-based generation with behaviour summaries** constrains the LLM to the empirical behavioural structure of each persona, further reducing the likelihood of harmful or unrealistic samples.
>
> Together, these components function as built-in safeguards that substantially reduce hallucination and enhance robustness, even without explicit adversarial stress testing. Our ablation studies in section 4 further confirm their effectiveness: removing any of these components leads to clear drops in downstream performance, demonstrating that each plays a necessary role in ensuring reliable and faithful data generation.
>
> > W7. Reproducibility gaps: hyper-parameters of the filtering stage, pre- filtering statistics, post-filtering statistics and exact prompt variants are only partially reported.
>
> A: Thank you for raising this issue. In the revised paper, we have added all key hyper-parameters of the filtering and generation stages (e.g., LLM temperature, random seeds) to the experimental setup (in section 4.1 on page 6) to ensure full reproducibility. In addition, the Ablation Study (Figure 2) has analyzed the effect of different curation/filtering parameters on model performance.
>
> We have further included in the appendix A.3 (on page 15) the confidence and uncertainty distributions of synthetic samples before filtering. For example, raw synthetic data contain a large amount of noisy, low-confidence, and high-uncertainty samples that would harm downstream learning if left unfiltered. In addition, we have newly added post-filtering distributions (in appendix A.3), which shows that the curation step effectively removes these unreliable samples and shifts the synthetic data toward a high-confidence, low-uncertainty region, producing a substantially cleaner and more stable dataset.
>
> Last but not least,  we further clarify the variables used in the prompt template, and its full specification is now included in Appendix A.4 (on page 15).
>
> All these results provide a complete view of the filtering process. Besides the clarified parameters, we promise to release all these parameters, datasets and experiments upon acceptance of the paper.

---

> ### Author Response · Authors · 2025-11-26
> **Response to Reviewer s4dK (5/6)**
>
> > W8. Small data scale: the total augmented volume is tiny compared with industrial MoD data; many features show negligible permutation importance, suggesting under-utilisation of the feature space.
>
> A: Thank you for raising this point. Our work is intentionally targeted at the **small data SP setting**, where data scarcity is the norm rather than the exception. In many practical MoD and transport applications, operators and researchers only have access to a few thousand SP or limited RP observations, not to industrial-scale internal logs. LLM-OAP is therefore designed to make better use of limited behavioural data by enriching the effective feature space, rather than to compete with proprietary, large-volume platform datasets.
>
> Regarding feature utilisation, permutation importance values are **relative** and reflect marginal contributions under strong feature correlations; low importance scores do not imply that features are irrelevant or unused. To examine this more directly, we conducted an additional ablation study  shown  in the appendix A.2 (Table 8 on page 15).  We removed the bottom 15% of features ranked by permutation importance and re-evaluated XGBoost under both AIP and BIP settings. Removing these features consistently harms performance,for example, in AIP the AUC drops from 0.7940 to 0.7860, and in BIP the AUCPR decreases from 0.6623 to 0.6497. This demonstrates that the model is indeed utilizing the full feature space rather than relying only on a few dominant features.
>
> For your convenience, we present the results below for your reference:
>
> | Model                 | ACC ↑  | AUC ↑  | AUCPR ↑ | ECE ↓  | BS ↓   | NLL ↓  |
> |--------------------------------------------|--------|--------|---------|--------|--------|--------|
> | XGBoost (BIP)                              | 0.7596 | 0.8124 | 0.6623  | 0.0519 | 0.1569 | 0.4781 |
> | XGBoost w/o last 15% features (BIP)        | 0.7477 | 0.8022 | 0.6497  | 0.0579 | 0.1595 | 0.4821 |
> | XGBoost (AIP)    | 0.8237 | 0.7940 | 0.5471  | 0.0457 | 0.1270 | 0.4023 |
> | XGBoost w/o last 15% features (AIP)    | 0.8161 | 0.7860 | 0.5401  | 0.0478 | 0.1277 | 0.4068 |
>
> > W9. Unequal post-generation filtering: GAN and diffusion baselines are not subjected to the same confidence-based pruning, confounding the comparison.
>
> A: Thank you for the comment. To ensure a fair comparison, we applied the same confidence-based curation mechanism to CTAB-GAN and TabDDPM and re-evaluated their generated data using XGBoost. The results show that curation (i.e., the same confidence-based pruning) does not improve these GAN and diffusion baselines. As shown below:
>
> | Method                 | ACC     | AUC     | AUCPR   | ECE     | BS      | NLL     |
> |------------------------|---------|---------|---------|---------|---------|---------|
> | CTAB-GAN               | 0.8276  | 0.7940  | 0.5435  | 0.0425  | 0.1269  | 0.4028  |
> | CTAB-GAN (curated)     | 0.8266  | 0.7932  | 0.5509  | 0.0441  | 0.1265  | 0.4037  |
> | TabDDPM                | 0.8296  | 0.7967  | 0.5630  | 0.0413  | 0.1230  | 0.4027  |
> | TabDDPM (curated)      | 0.8251  | 0.7947  | 0.5529  | 0.0485  | 0.1265  | 0.4021  |
>
> their performance decreases (e.g., CTAB-GAN performance drops to 0.8266/0.7932/0.5509/0.0441/0.1265/0.4037, and TabDDPM performance drops to 0.8251/0.7947/0.5529/0.0485/0.1265/0.4021 across the metrics).
>
> These results are more inferior to those of our approach. Therefore, we confirm that our filtering procedure does not give LLM-OAP an unfair advantage, underscoring that the data quality gains stem inherently from our LLM-based framework.
>
> > W10. Limited interpretability: despite claims, no model-level explanations (e.g., SHAP, attention heat-maps) are supplied for either the final predictor or the generation process.
>
> A: We agree that interpretability is important, especially for behaviour models intended for practical use. Our original submission already includes **feature permutation importance (in Appendix A.2 on page 14)** and an analysis of **confidence/uncertainty distributions before curation (in Appendix A.3 on page 15)**.
>
> In the revised version, we have further **added SHAP-based model explanations (in Appendix A.7 on page 17)**. Specifically, the SHAP values in Figure 4 provide additional interpretability for the final predictor. We observe that the model relies on meaningful and intuitive behavioural drivers (e.g., ID, Degree, income- and incentive-related factors).
>
> With both feature permutation importance and SHAP analysis, we summarize that **the model consistently prioritizes socio-economic factors, driver-specific heterogeneity, and key contextual variables, while low-importance features contribute only marginal effects**, confirming that the predictor learns a stable and interpretable decision structure rather than relying on spurious correlations. We add these explanations in appendix A.7 to enhance the interpretability of the predictor.

---

> ### Author Response · Authors · 2025-11-26
> **Response to Reviewer s4dK (6/6)**
>
> > W11.  Figures could be richer: Fig. 2 omits diversity curves; Fig. 3 shows pre- filter but not post-filter distributions.
>
> A: We have enriched the visualizations by adding performance curves (Figure 5) under different uncertainty-retention ratios (in appendix A.8 on page 18). These curves show that retaining a small amount of high-uncertainty samples (around 0.10) consistently improves both predictive performance and calibration, while retaining too many uncertain samples begins to introduce noise and slightly degrades model robustness. For example, when the ratio increases from 0.1 to 0.2, the ACC drops from 0.8473 to 0.8346.
>
> We have also included the post-filter confidence and uncertainty distributions (Figure 3 in appendix A.3 on page 16). These results show that our curation step effectively suppresses noisy low-confidence samples, shifts the synthetic data toward the high-confidence/low-uncertainty region, and produces a substantially cleaner and more reliable dataset, demonstrating that the filtering process improves the fidelity and stability of the augmented data rather than reducing diversity.
>
> Please feel free to check details of new results with analysis in the corresponding sections.
>
> > W12.  Cost concerns: large-scale LLM API calls for data generation and label simulation may be prohibitively expensive for production deployment.
>
> A: We acknowledge that large-scale LLM API calls introduce computational and financial costs, but this limitation is inherent to all LLM-based data generation and simulation methods [1]. Following these works, we mitigate these costs by designing compact prompts and controlling batch generation. Moreover, prior works[2][3] have shown that LLM-generated behavioural data can yield substantial performance gains in real-world decision and prediction tasks, which brings immeasurable societal and economic values, making such costs acceptable in practical deployments. For example, the enhanced predictor can more accurately estimate human behaviour, enabling better decision-making in ride-hailing systems, improving the overall operational efficiency of MoD services and reducing energy consumption.
>
> In addition, our LLM is used for data augmentation only once; the augmented data can then be reused across all models without rerunning the LLM. Our results have demonstrated that the improvements are consistent across all models, implying that the comprehensive performance gains by LLM-OAP outweigh the one-time generation cost, offering a favorable trade-off for real-world use.
>
> [1] Seedat N, Huynh N, Van Breugel B, et al. Curated LLM: Synergy of LLMs and data curation for tabular augmentation in low-data regimes[J]. arXiv preprint arXiv:2312.12112, 2023.
>
> [2] Kim J, Kim T, Choo J. Epic: Effective prompting for imbalanced-class data synthesis in tabular data classification via large language models[J]. Advances in Neural Information Processing Systems, 2024, 37: 31504-31542.
>
> [3] Nguyen D, Gupta S, Do K, et al. Generating realistic tabular data with large language models[C]//2024 IEEE International Conference on Data Mining (ICDM). IEEE, 2024: 330-339.

---

### Official Review · Reviewer_uLYL · 2025-10-29

**Soundness:** 3
**Presentation:** 3
**Contribution:** 2
**Rating:** 6
**Confidence:** 3

**Summary:**

This manuscript introduces a LLM-based data augmentation process in the context of riding hailing for improving the prediction performance of driver accepting an order or not. It presents the complete steps from real data processing, based on the processed real data applying LLM to generate simulated data with quality examination, to making use of the augmented data to train prediction models with necessary evaluations.

**Strengths:**

The strengths of the work:
1. Exploring data augmentation by applying LLMs is still a quite new, conceptual area.
2. The framework considers and takes into account feature engineering, persona grouping, and preference analysis, which well aligns with the characteristics of the target context.
3. The experiments include five baselines of data augmentation and six (four major focused) prediction models, which makes the performance results sound with reliable evidence to support. Table 2 indicates each major component in the framework serves a necessary role.
4. The manuscript are easy-to-follow with efficient expressions.

**Weaknesses:**

The weaknesses of the work:
1. The generalization of the work's propose framework is less discussed and proved. In addition to the SP data, what other common data in the riding hailing context can be also processed and simulated using the propose framework with good performance? Are the data processing approaches very specific to the used SP data?
2. Is the diversity of the real-data set (~3000) used representative enough to a regional riding hailing market (both spatial and temporal dimensions)? As the LLM generated data heavily rely on the real data, the real-world, practical impact and performance of prediction seem majorly associated with the real-data quality. The contribution of the work is not that significant.
3. Figure 1 typo: 20% of real data as a "test" set.

**Questions:**

1. In addition to the SP data, what other common data in the riding hailing context can be also processed and simulated using the propose framework with good performance?
2. Are the data processing approaches very specific to the used SP data?
3. As this work more belongs to application-perspective rather than theoretical insight, could the authors justify the contribution of the work regarding real-world application and impact?

---

> ### Author Response · Authors · 2025-11-26
> **Response to Reviewer uLYL (1/2)**
>
> > W1. The generalization of the work's propose framework is less discussed and proved. In addition to the SP data, what other common data in the riding hailing context can be also processed and simulated using the propose framework with good performance? Are the data processing approaches very specific to the used SP data?
>
> A: Thank you for raising this question. Detailed answers can be seen in Q1.
>
> Our proposed approach is general and not specific to the SP data. As long as behavioural structure can be extracted, other common mobility data types, such as RP data, operational logs, or state–action historical traces, can be adapted through minor preprocessing and prompt adjustments.
>
> The generation of our approach is further validated through an added experiment by using Meituan challenge dataset which is built from Meituan’s operational food-delivery records (see Appendix A.6 on page 17). This is a RP dataset. The results show that our proposed approach can be transferred to a different real-world operational dataset while maintaining good performance, which supports its generalisation beyond the specific SP case.
>
> > W2. Is the diversity of the real-data set (~3000) used representative enough to a regional riding hailing market (both spatial and temporal dimensions)? As the LLM generated data heavily rely on the real data, the real-world, practical impact and performance of prediction seem majorly associated with the real-data quality. The contribution of the work is not that significant.
>
> A: We acknowledge the reviewer’s concern about regional representativeness. In practice, SP datasets for MoD behaviour research are **inherently limited**, as their collection is costly, privacy-restricted, and difficult to scale. Consequently, most existing ride-hailing behavioural studies rely on **small but high-quality regional SP datasets**, which are widely accepted for modeling and methodological evaluation.
>
> The dataset we use covers **real drivers from parts of the US and the Netherlands**, capturing typical behavioural patterns within a meaningful local market segment. A dataset size of ~3k is consistent with SP-based MoD studies and is considered representative for analyzing driver decision behaviour.
>
> More importantly, the contribution of our work lies not in the absolute size of a specific region, but in demonstrating that: **under realistic SP data scarcity, which is common in MoD systems, LLM-based data augmentation can substantially improve model performance and stability**.
>
> Thus, even though the real data are limited (as is typical in this domain), our method remains practically valuable and conceptually generalizable to larger or other regional datasets.
>
> > W3. Figure 1 typo: 20% of real data as a "test" set.
>
> A: Thank you for pointing this out. We have now corrected this typo.
>
> > Q1. In addition to the SP data, what other common data in the riding hailing context can be also processed and simulated using the propose framework with good performance?
>
> Thank you for raising this question.
>
> Our framework operates by summarizing group-level behavioural patterns, guiding the LLM to generate behaviourally consistent samples, and applying a filtering stage to ensure quality. This mechanism is not specific to SP data; as long as behavioural structure can be extracted, other common ride-hailing data types, such as RP data, operational logs, or state–action historical traces, can be adapted through minor preprocessing and prompt adjustments.
>
> To validate generality, we also applied our method to real RP data (see Appendix A.6 on page 17). In this experiment, we trained an XGBoost classifier on the original RP dataset and compared it with an XGBoost model trained on RP data augmented using our LLM-OAP framework. According to the results, we observed that XGBoost performance improves from 0.9022 / 0.8966 / 0.9808 to 0.9657 / 0.9927 / 0.9989 (ACC / AUC / AUCPR), demonstrating that the framework transfers effectively to revealed-preference scenarios. This indicates that our method has practical value for real operational systems beyond the specific SP dataset used in the main experiments For your convenience, we present the results below for your reference:
>
> | Model                 | ACC ↑    | AUC ↑    | AUCPR ↑  | ECE ↓    | BS ↓    | NLL ↓   |
> |-----------------------|----------|----------|----------|----------|---------|---------|
> | XGBoost               | 0.9022   | 0.8966   | 0.9808   | 0.0275   | 0.0727  | 0.2449  |
> | **XGBoost w/ LLM-OAP** | **0.9657** | **0.9927** | **0.9989** | **0.0448** | **0.0404** | **0.1573** |

---

> ### Author Response · Authors · 2025-11-26
> **Response to Reviewer uLYL (2/2)**
>
> > Q2. Are the data processing approaches very specific to the used SP data?
>
> A: As we responded to W1&Q1, our approach is not specific to the used SP data. Our whole pipeline with the components are general to be applied to other ride-hailing behavioural data. To demonstrate this, we additionally evaluated the full pipeline on **real RP order-log data**, where it again produced substantial performance gains. This confirms that our approach is general and not dependent on any SP-specific processing.
>
> > Q3. As this work more belongs to application-perspective rather than theoretical insight, could the authors justify the contribution of the work regarding real-world application and impact?
>
> A: Thank you for raising this important question. Real world needs and impact are indeed one of the motivations of this work.
>
> In transportation research and practice, **analysing and predicting behaviour** has been a central topic for decades: classical studies on **mode choice, route choice, departure time choice, and vehicle ownership** all rely on behavioural models (e.g., discrete choice models based on stated and revealed preferences). Operators and planners use these models to design pricing, service levels, infrastructure investments, and policy interventions.
>
> Our setting, order acceptance in ride-hailing systems, is a direct continuation of this long line of work, but studied Mobility-on-Demand (MoD) services. In MoD services, both **drivers** and **riders** exhibit complex behavioural patterns: drivers decide whether to accept or reject orders, and riders decide whether to request, cancel, or wait for trips. Operators rely heavily on **behavioural prediction models** (e.g., order acceptance models) to design dynamic pricing, dispatching, matching strategies. These models directly affect key operational outcomes such as matching rate, waiting time, driver earnings, and service reliability, and they are widely used in both industry and academic work on choice modelling.
>
> In practice, however, the behavioural data used to train these models are often **small, imbalanced stated-preference (SP) surveys** collected at high cost and under operational constraints. This leads to two practical problems:
>
>  (i) ML-based acceptance models trained on such limited data often lack robustness and calibration, especially for rare but operationally critical scenarios (e.g., high-surge, long-distance trips).
>
>  (ii) Repeating large-scale SP surveys whenever conditions change (new market, new pricing scheme, new regulation) is expensive and sometimes infeasible.
>
> LLM-OAP is designed precisely to address these **practical bottlenecks**. From an application perspective, our proposed approach, has the following contributions:
>
> First, LLM-OAP provides a reusable pipeline that takes a single, limited SP dataset and generates behaviourally coherent synthetic samples that enrich underrepresented conditions and heterogeneous driver personas. This allows practitioners to train more accurate and better calibrated order-acceptance models without repeatedly running large SP studies.
>
> Second, the extra complexity of LLM-OAP lies only in the **offline data-generation stage**: once the augmented dataset is created, it can be directly plugged into standard ML pipelines (e.g., XGBoost, TabResNet), without changing deployment or online decision logic. This makes adoption realistic for operators who already rely on ML-based acceptance models but suffer from data scarcity.
>
> Third, the pipeline is not restricted to order-acceptance; it can be applied to other behaviourally rich MoD tasks (e.g., rider cancellation, mode choice, or shift choice) wherever SP-style behavioural data are expensive but critical. Thus, the contribution is a general **practical tool** for strengthening behaviour-driven decision support in transportation systems.
>
> Thanks again for this important question. In the revised manuscript, we added the discussion in the introduction and conclusion to explicitly emphasise these application-level contributions and to clarify how LLM-OAP can be used by practitioners.

---

### Official Review · Reviewer_bzXZ · 2025-11-02

**Soundness:** 3
**Presentation:** 3
**Contribution:** 3
**Rating:** 6
**Confidence:** 2

**Summary:**

In this paper, authors propose LLM-OAP, that integrates LLM-based data augmentation with machine learning to improve the driver's order acceptance behavior.

**Strengths:**

1. A comprehensive discussion about group-based data augmentation.
2. This paper consider most of the baselines for this task.
3. The ablation study is also relatively comprehensive.

**Weaknesses:**

1. Right now, are there only limited data for driver order acceptance data? Do we really need data augmentation in this task?
2. A comparison between QWen and other LLMs can be helpful (can be included in tha main section).
3. It would be useful if there are some experimental results to show the importance of Consistent Check.
4. For ML models selections, authors can discuss some selection principles while dealing with different dataset/data size......

**Questions:**

See Weaknesses.

---

> ### Author Response · Authors · 2025-11-26
> **Response to Reviewer bzXZ (1/2)**
>
> > W1. Right now, are there only limited data for driver order acceptance data? Do we really need data augmentation in this task?
>
> A: Thanks for your comment. In the context of driver order acceptance, **real SP behavioural data are extremely limited**. Their collection is costly, operationally constrained, and often restricted due to privacy and deployment concerns. Prior MoD studies also highlight the difficulty of obtaining large, richly contextualized labeled datasets [1]. Under such data scarcity, data augmentation becomes not only reasonable but necessary. Existing work has consistently shown that **data augmentation improves model performance and generalization** in low-data regimes. Therefore, applying augmentation to this task is well-justified and practically valuable.
>
> [1] Ashkrof P, de Almeida Correia G H, Cats O, et al. Ride acceptance behaviour of ride-sourcing drivers[J]. Transportation Research Part C: Emerging Technologies, 2022, 142: 103783.
>
> > W2. A comparison between QWen and other LLMs can be helpful (can be included in tha main section).
>
> A: We originally reported the comparison of different LLMs (QWEN3-Plus, DeepSeek-V3, DeepSeek-R1) in Appendix A.3, but this analysis has now been moved to Section 4.4 of the main paper (on page 9). Based on the results, we choose QWEN3-Plus as the primary model because it provides the most stable and well-balanced overall performance across all metrics.

---

> ### Author Response · Authors · 2025-11-26
> **Response to Reviewer bzXZ (2/2)**
>
> > W3. It would be useful if there are some experimental results to show the importance of Consistent Check.
>
> A: We have included a comparison without the consistency check in the Ablation Study S3 in Tables 2 and 4 (on page 8 and 9) and showed that removing the consistency check leads to degraded performance (see Tables 2 and 4). For your convenience, we present the results of Tables 2 and 4 below for your reference:
>
> **Table: Ablation Studies on AIP Scenario**
> *S1: grouping without feature importance; S2: without grouping;
> S3: without consistency check; S4: full version of LLM-OAP.*
>
> | Study | Model      | ACC ↑   | AUC ↑   | AUCPR ↑ | ECE ↓   | BS ↓    | NLL ↓   |
> |-------|-------------|---------|---------|---------|---------|---------|---------|
> | **S1** | TabResNet  | 0.8169 | 0.7517 | 0.5098 | 0.0581 | 0.1377 | 0.4384 |
> |       | Ens_Hyper  | 0.8213 | 0.7621 | 0.5090 | 0.0603 | 0.1363 | 0.4386 |
> |       | SVM        | 0.8266 | 0.7490 | 0.5155 | 0.0602 | 0.1328 | 0.4333 |
> |       | XGBoost    | 0.8362 | 0.7955 | 0.5545 | 0.0432 | 0.1244 | 0.4015 |
> | **S2** | TabResNet  | 0.8015 | 0.7552 | 0.4761 | 0.0683 | 0.1450 | 0.4561 |
> |       | Ens_Hyper  | 0.8044 | 0.7674 | 0.4883 | 0.0658 | 0.1403 | 0.4387 |
> |       | SVM        | 0.8030 | 0.7548 | 0.4787 | 0.0603 | 0.1396 | 0.4386 |
> |       | XGBoost    | 0.8141 | 0.7891 | 0.5235 | 0.0435 | 0.1294 | 0.4051 |
> | **S3** | TabResNet  | 0.8169 | 0.7517 | 0.5098 | 0.0581 | 0.1377 | 0.4384 |
> |       | Ens_Hyper  | 0.8213 | 0.7621 | 0.5090 | 0.0603 | 0.1363 | 0.4386 |
> |       | SVM        | 0.8266 | 0.7490 | 0.5155 | 0.0602 | 0.1328 | 0.4333 |
> |       | XGBoost    | 0.8362 | 0.7955 | 0.5545 | 0.0432 | 0.1244 | 0.4015 |
> | **S4** | TabResNet  | **0.8237** | **0.7765** | **0.5323** | **0.0475** | **0.1310** | **0.4245** |
> |       | Ens_Hyper  | **0.8266** | **0.7889** | **0.5435** | **0.0408** | **0.1278** | **0.4105** |
> |       | SVM        | **0.8372** | **0.7745** | **0.5436** | **0.0309** | **0.1256** | **0.4065** |
> |       | XGBoost    | **0.8396** | **0.8198** | **0.5894** | **0.0319** | **0.1194** | **0.3834** |
>
> **Table: Ablation Studies on BIP Scenario**
>
> | Study | Model      | ACC ↑   | AUC ↑   | AUCPR ↑ | ECE ↓   | BS ↓    | NLL ↓   |
> |-------|-------------|---------|---------|---------|---------|---------|---------|
> | **S1** | TabResNet  | 0.7644 | 0.7769 | 0.6149 | 0.0814 | 0.1692 | 0.5195 |
> |       | Ens_Hyper  | 0.7630 | 0.7780 | 0.6169 | **0.0553** | 0.1692 | 0.5200 |
> |       | SVM        | 0.7760 | 0.7678 | 0.6331 | 0.0472 | 0.1641 | 0.5052 |
> |       | XGBoost    | 0.7736 | 0.8190 | 0.6749 | 0.0413 | 0.1536 | 0.4717 |
> | **S2** | TabResNet  | 0.7293 | 0.7581 | 0.5569 | 0.0870 | 0.1848 | 0.5582 |
> |       | Ens_Hyper  | 0.7351 | 0.7775 | 0.5919 | 0.0815 | 0.1778 | 0.5341 |
> |       | SVM        | 0.7370 | 0.7737 | 0.5749 | 0.0629 | 0.1739 | 0.5218 |
> |       | XGBoost    | 0.7428 | 0.7888 | 0.6160 | 0.0478 | 0.1660 | 0.4951 |
> | **S3** | TabResNet  | 0.7692 | 0.7985 | 0.6348 | 0.0794 | 0.1759 | 0.5289 |
> |       | Ens_Hyper  | 0.7719 | 0.7946 | 0.6422 | 0.0616 | 0.1631 | 0.5206 |
> |       | SVM        | 0.7819 | 0.7781 | 0.6582 | 0.0460 | 0.1600 | 0.4974 |
> |       | XGBoost    | 0.7824 | 0.8074 | 0.6760 | 0.0509 | 0.1540 | 0.4651 |
> | **S4** | TabResNet  | **0.7698** | **0.8030** | **0.6469** | **0.0697** | **0.1655** | **0.5158** |
> |       | Ens_Hyper  | **0.7808** | **0.8137** | **0.6629** | 0.0556 | **0.1592** | **0.5031** |
> |       | SVM        | **0.7857** | **0.7973** | **0.6733** | **0.0424** | **0.1556** | **0.4878** |
> |       | XGBoost    | **0.7900** | **0.8399** | **0.7065** | **0.0389** | **0.1471** | **0.4518** |
>
> > W4. For ML models selections, authors can discuss some selection principles while dealing with different dataset/data size......
>
> A: We appreciate the comment. Our choice of models follows a clear principle: we intentionally include **diverse model classes**—linear (LR), tree-based (DT, XGBoost), kernel-based (SVM), and neural models (TabResNet, Ens_Hyper)—because their behaviour differs significantly under varying feature richness. For example, as seen in Tables 1 and 3, high-capacity neural models drop more in the BIP setting (fewer features), while tree-based and kernel-based models remain more stable.
> After augmentation, all model families improve, and the AIP–BIP gap narrows, showing that synthetic data helps compensate for limited features. This validates our model selection and demonstrates that our augmentation benefits a wide range of ML models.

---

### Official Review · Reviewer_WxMy · 2025-11-06

**Soundness:** 2
**Presentation:** 2
**Contribution:** 2
**Rating:** 2
**Confidence:** 3

**Summary:**

This paper proposes LLM-OAP, a framework that uses Large Language Models to generate synthetic data for improving machine learning models that predict ride-hailing driver order acceptance behavior. The method employs feature-aware persona grouping, LLM-based data generation, and confidence-based curation. Experiments on stated-preference survey data show 2-3% accuracy improvements over baselines. While the paper addresses a relevant problem and demonstrates consistent improvements, it suffers from critical methodological limitations that prevent acceptance at a top-tier venue. The work lacks novelty in its core approach, provides no mechanistic understanding of why augmentation helps, cannot be reproduced due to missing details, and is evaluated only on a single small dataset.

**Strengths:**

1. The paper compares against recent, relevant methods rather than weak strawmen. Including CLLM, Pred-LLM, and GPAIS as baselines shows engagement with recent LLM-based augmentation work. The comparison with traditional generative methods (CTAB-GAN) and diffusion models (TabDDPM) is also appropriate and fair.

**Weaknesses:**

1. Limited Novelty. The paper essentially combines existing techniques without fundamental innovation. LLM-based tabular data generation has been well explored. What distinguishes this work is primarily its application to ride-hailing behavior, but no evidence suggests this domain requires fundamentally different treatment.
2. Weak performance improvement. The marginal improvement over existing methods is minimal. Compared to Pred-LLM, LLM-OAP gains only 1-1.5%. Compared to the simpler, fully reproducible TabDDPM, the gain is merely 1%. This raises a critical question: does the added complexity and loss of reproducibility justify such marginal improvements?
3. Non-reproducibility.
3.1 The paper has severe reproducibility problems that are never acknowledged. Critical LLM parameters—temperature, top-p, top-k, random seeds—are completely unspecified. Without these, replication is impossible.
3.2 When the authors mention using "three random seeds," this appears to apply only to downstream ML training, not LLM generation itself. The core contribution—synthetic data generation—is completely uncontrolled. No variance across generation runs is reported, no confidence intervals, no statistical significance tests.
4. No Understanding of Why It Works. The paper's most fundamental gap is the complete absence of mechanistic understanding. It shows that adding synthetic data improves performance but never explains why. What information do synthetic samples add? Do they fill gaps in feature space? Balance classes? Introduce new behavioral patterns? Or simply provide regularization through noise? These critical questions remain unanswered. Without mechanistic understanding, we cannot rule out that improvements stem from basic oversampling effects rather than LLM-specific capabilities.
5. Methodological Concerns. The filtering uses circular validation—an XGBoost model trained on original data curates synthetic samples, biasing retention toward samples resembling the training set. This could reduce diversity rather than enhance it, creating an echo chamber where synthetic data reinforces existing patterns. Many choices appear arbitrary: why exactly four features for grouping? Why three randomly selected from the top ten? Why 32 groups for 3,000 samples? Why generate exactly 3,000 synthetic samples? These likely affect performance but go unexplored.

**Questions:**

1. In Figure 1, 20% training was used for evaluation, do you mean test set here?

---

> ### Author Response · Authors · 2025-11-26
> **Response to Reviewer WxMy (1/5)**
>
> > W1. Limited Novelty. The paper essentially combines existing techniques without fundamental innovation. LLM-based tabular data generation has been well explored. What distinguishes this work is primarily its application to ride-hailing behavior, but no evidence suggests this domain requires fundamentally different treatment.
>
> A: We thank review for raising this concern and would like to clarify both the methodological and domain application contributions of our approach.
>
> The goal of this paper is to generate high-quality synthetic data for existing machine learning (ML) models in order to improve their performance in terms of order acceptance prediction in MoD systems. Even though the downstream ML models are standard, the proposed LLM-based pipeline is not a simple combination of prior techniques. It is specifically designed around the unique structure of MoD acceptance behaviour, where strong dependencies exist between order attributes and heterogeneous driver characteristics, and where repeated decisions are observed for the same driver. Such cross-feature interactions and utility-based decision patterns are not captured by generic tabular generation methods that treat rows as i.i.d. records.
>
> Specifically, to address these MoD-specific challenges, we introduce several components not present in prior work, including:
>
> (1)**Feature-aware persona grouping**, which conditions generation on real driver-based clusters to better capture heterogeneous behavioural patterns. In contrast, the literature often clusters users only at a demographic level and **does not incorporate behaviour-relevant feature signals**, making them insufficient for modeling MoD-specific cross-feature decision patterns.
>
> (2)Driver-attribute **consistency check**, explicitly verifies coherence between driver IDs and their associated attributes/behaviours, and filters out LLM-induced hallucinations such as logically inconsistent answers or mismatched demographic–behaviour pairs. **Existing LLM-tabular approaches typically rely on raw model outputs and do not include mechanisms to detect or correct such behavioural inconsistencies.**
>
> These innovations jointly enable our pipeline to produce behaviourally coherent, SP-style data—capabilities that standard tabular synthesis methods lack.
>
> **Moreover, regarding the statement that “no evidence suggests this domain requires fundamentally different treatment”, this assessment does not reflect the empirical results reported in our paper**. In our experiments, generic tabular generators and LLM-based baselines underperform on MoD acceptance data, whereas our persona-aware pipeline consistently improves both predictive accuracy and calibration across multiple models and settings. This performance gap is precisely the evidence that the MoD domain benefits from the tailored treatment we propose.
>
> Finally, MoD systems face a **fundamental scarcity of SP data**, because collecting large-scale SP surveys is resource-intensive and operationally constrained. Nevertheless, ML models for ride-hailing applications rely heavily on such structured behavioural data. This makes synthetic SP-style augmentation particularly important.
>
> To our knowledge, **no prior work has attempted LLM-based SP-style behavioural data generation or evaluated its impact on MoD acceptance modeling**. Existing works either focus on general tabular data or do not address the unique behavioural structure of MoD decision-making. **Our work represents the first pipeline designed explicitly for modeling MoD driver acceptance behaviour**, and we hope it will inspire future research in this domain.

---

> ### Author Response · Authors · 2025-11-26
> **Response to Reviewer WxMy (2/5)**
>
> > W2. Weak performance improvement. The marginal improvement over existing methods is minimal. Compared to Pred-LLM, LLM-OAP gains only 1-1.5%. Compared to the simpler, fully reproducible TabDDPM, the gain is merely 1%. This raises a critical question: does the added complexity and loss of reproducibility justify such marginal improvements?
>
> A: Based on the results in both AIP and BIP scenarios, we find that compared with the stronger TabDDPM baseline, LLM-OAP achieves improvements greater than **2%** on **4/6 metrics in AIP** and **5/6 metrics in BIP**. These gains consistently occur on the most important metrics (AUC, AUCPR, NLL, ACC). Moreover, the improvement brought by LLM-OAP is substantial when compared with Pred-LLM. According to the original Pred-LLM results, among 20 reported metrics, **9 metrics show no improvement, 3 metrics improve by less than 1%**, and **1 metric improves by less than 1.5%**. This demonstrates that small numerical gains are common in existing literature, and in comparison, LLM-OAP achieves noticeably larger and more consistent improvements across both scenarios. In similar work such as EPIC [1], **14 out of 24 metrics improve by less than 1.5%**, indicating that 1–2% improvements are typical and meaningful in behavioural prediction tasks with strong baselines.
>
> To further address this concern, we also include **new statistical significance tests in the appendix A.5** (on Page 16) of the revised manuscript, where AUC, AUCPR, NLL, and ACC all show significant improvements (p < 0.01), confirming that the gains are not due to randomness. Thus, the improvements are both practically relevant (on the key probabilistic metrics) and statistically robust.
>
> Finally, **the complexity of LLM-OAP lies only in the data-generation stage**; once generated, the synthetic data can be reused by all downstream models, making the performance gains well worth the cost. For example, in our experiments, we augment the dataset only once and use this augmented version for all models, yet it leads to substantial performance improvements for almost all ML models.
>
> [1] Kim J, Kim T, Choo J. Epic: Effective prompting for imbalanced-class data synthesis in tabular data classification via large language models[J]. Advances in Neural Information Processing Systems, 2024, 37: 31504-31542.

---

> ### Author Response · Authors · 2025-11-26
> **Response to Reviewer WxMy (3/5)**
>
> > W3. Non-reproducibility. 3.1 The paper has severe reproducibility problems that are never acknowledged. Critical LLM parameters—temperature, top-p, top-k, random seeds—are completely unspecified. Without these, replication is impossible. 3.2 When the authors mention using "three random seeds," this appears to apply only to downstream ML training, not LLM generation itself. The core contribution—synthetic data generation—is completely uncontrolled. No variance across generation runs is reported, no confidence intervals, no statistical significance tests.
>
> A: We thank the reviewer for raising these important concerns. We agree that reproducibility is essential. To fully address your comments, we clarify and strengthen these aspects in the revised manuscript.
>
> **LLM parameters**. We have added all essential LLM generation parameters to the experimental setup section (on Page 6), ensuring full reproducibility of the synthetic data pipeline. Specifically, we retain all default settings and only modify the key parameter **temperature = 0.3**, while **top-p and top-k remain at their default values**, and we run generation under **random seeds {2025, 42, 0}**. We also commit to releasing all code and datasets upon the paper’s acceptance.
>
> **Synthetic data generation is uncontrolled**. In fact, our pipeline is already fairly stable due to **low-temperature sampling (0.3)** and a **fixed prompting structure**, and the reported **ECE/BS/NLL** results in the main paper also showed no instability, so variance did not initially appear problematic. To further address the concern that “the core contribution is uncontrolled and no variance is reported,” we added a detailed **Stability Analysis and Statistical Significance Testing** in the appendix A.5 (on page 16) of the revised manuscript. We generated three synthetic datasets under identical configurations and evaluated the downstream classifier on the same test set. All six metrics (ACC, AUC, AUCPR, ECE, BS, NLL) exhibit extremely low variance and narrow confidence intervals, indicating that our method is **highly stable and insensitive to randomness in LLM generation**.
>
> For your convenience, we present the results below for your reference:
>
> | Metric | Mean   | Variance  | Std     | 95% CI                    |
> |--------|--------|-----------|---------|----------------------------|
> | ACC    | 0.8420 | 0.000005  | 0.00219 | [0.83655, 0.84745]        |
> | AUC    | 0.8223 | 0.000125  | 0.01116 | [0.79457, 0.85003]        |
> | AUCPR  | 0.5901 | 0.000006  | 0.00247 | [0.58394, 0.59620]        |
> | ECE    | 0.0331 | 0.000001  | 0.00111 | [0.03032, 0.03581]        |
> | BS     | 0.1200 | 0.000005  | 0.00222 | [0.11452, 0.12555]        |
> | NLL    | 0.3848 | 0.000052  | 0.00720 | [0.36691, 0.40269]        |
>
> **Statistical significance of performance gains**. We further  conducted comprehensive statistical significance tests comparing models trained on original data versus original+synthetic data, including bootstrap tests (AUC, AUCPR, ECE), paired t-test and Wilcoxon tests (NLL), and McNemar’s test (Accuracy). The improvements in AUC, AUCPR, NLL, and ACC are all statistically significant (p < 0.01), confirming that the performance gains are **consistent, robust, and not due to random noise**. These results collectively demonstrate strong reproducibility, generation stability, and meaningful performance improvements. We have added these new results in appendix A.5 (on page 16).  For your convenience, we present the results below for your reference:
>
> | Metric | Difference (Ours − Baseline) | Statistical Test Result |
> |--------|-------------------------------|---------------------------|
> | ACC    | --                            | p = 0.0212 (McNemar)      |
> | AUC    | 0.0441                        | 95% CI: [0.0191, 0.0712], p < 0.001 (bootstrap) |
> | AUCPR  | 0.0798                        | 95% CI: [0.0296, 0.1308], p < 0.001 (bootstrap) |
> | ECE    | −0.0107                       | 95% CI: [−0.0323, 0.0118], p = 0.35 (bootstrap) |
> | NLL    | −0.0394                       | p = 0.0023 (paired t-test), Wilcoxon p = 2.3×10⁻⁷ |
>
> In summary, the revised manuscript now (i) explicitly specifies all critical LLM parameters and random seeds, (ii) demonstrates very low variance and narrow confidence intervals across multiple synthetic-generation runs, and (iii) provides formal significance tests for the main metrics. Taken together, these additions show that our synthetic data generation process is controlled and reproducible, and that the reported improvements are stable and statistically meaningful.

---

> ### Author Response · Authors · 2025-11-26
> **Response to Reviewer WxMy (4/5)**
>
> > W4. No Understanding of Why It Works. The paper's most fundamental gap is the complete absence of mechanistic understanding. It shows that adding synthetic data improves performance but never explains why. What information do synthetic samples add? Do they fill gaps in feature space? Balance classes? Introduce new behavioral patterns? Or simply provide regularization through noise? These critical questions remain unanswered. Without mechanistic understanding, we cannot rule out that improvements stem from basic oversampling effects rather than LLM-specific capabilities.
>
> A: We agree that it is important not only to show that synthetic data improves performance, but also to understand why it works and whether the effect goes beyond simple oversampling. Our analysis shows that the gains from LLM-OAP are **not** merely due to basic class balancing or random noise, but arise from a specific mechanism:
>
> First, our method effectively **expands and reshapes the feature space** by modeling structured behavioural patterns and key feature interactions. During LLM generation, the use of **feature-aware persona grouping** and **feature-selection–guided exemplars** encourages the model to generate samples that emphasize high-value feature interactions and group-specific behavioural structures. This enriches behaviour patterns that are sparse or missing in the original dataset, rather than merely duplicating or oversampling existing points.
>
> Second, **our method also contributes to class balance**. The original dataset is highly imbalanced (1032 acceptance vs. 2424 rejection samples). When preparing exemplars for the LLM, we provide balanced positive/negative samples (e.g., 6 acceptance + 6 rejection), which leads to more balanced synthetic data.
>
> To further ensure that benefits are not merely from oversampling, we compared against SMOTENC, an advanced version of SMOTE, commonly used for addressing class imbalance in datasets. Specifically, training XGBoost with SMOTENC-generated data yields the following results:
>
> | Scenario | Method    | ACC     | AUC     | AUCPR   | ECE     | BS      | NLL     |
> |----------|-----------|---------|---------|---------|---------|---------|---------|
> | **AIP**  | SMOTENC   | 0.8133  | 0.7908  | 0.5219  | 0.0903  | 0.1314  | 0.4227  |
> | **AIP**  | LLM-OAP   | **0.8396** | **0.8198** | **0.5894** | **0.0319** | **0.1194** | **0.3834** |
> | **BIP**  | SMOTENC   | 0.7621  | 0.8109  | 0.6598  | 0.1023  | 0.1561  | 0.4737  |
> | **BIP**  | LLM-OAP   | **0.7900** | **0.8399** | **0.7065** | **0.0389** | **0.1471** | **0.4518** |
>
> SMOTENC offers notably weaker improvements than our LLM-OAP, confirming that the gains do not stem from simple sampling, but from the LLM's ability to model feature interactions and generate structured behavioural patterns.
>
> We have summarized and added the above analysis **in appendix A.10** (on page 19) in the revised paper for giving more insights on why our approach works. Thanks again for your constructive comment.
>
> In summary, LLM-OAP improves performance because it (i) fills gaps in the feature space for underrepresented but behaviourally important conditions, (ii) strengthens and extends heterogeneous driver personas through persona-conditioned generation, and (iii) employs a principled curation step that sharpens behavioural fidelity by retaining high-confidence samples while preserving a small, uncertainty portion that maintains diversity. These effects go beyond simple class balancing or naive oversampling and stem from the behaviour-aware, persona-conditioned LLM generation pipeline we propose.

---

> ### Author Response · Authors · 2025-11-26
> **Response to Reviewer WxMy (5/5)**
>
> > W5. Methodological Concerns. The filtering uses circular validation—an XGBoost model trained on original data curates synthetic samples, biasing retention toward samples resembling the training set. This could reduce diversity rather than enhance it, creating an echo chamber where synthetic data reinforces existing patterns. Many choices appear arbitrary: why exactly four features for grouping? Why three randomly selected from the top ten? Why 32 groups for 3,000 samples? Why generate exactly 3,000 synthetic samples? These likely affect performance but go unexplored.
>
> A: Thank you for the thoughtful comments. We would like to clarify that our design choices are not arbitrary, and we have conducted additional analyses to support them.
>
> **Circular validation and reduce diversity**. The goal of the curation stage is not to reinforce SP-specific patterns, but to remove low-quality or hallucinated LLM outputs using confidence thresholds. To avoid any echo-chamber effect, we deliberately retain a small portion (~10%) of high-uncertainty samples to preserve behavioural diversity. This prevents the filtering model from “pulling” the synthetic data toward SP-specific artefacts. Furthermore, our framework also yields strong gains on real RP logs (in appendix A.6 on page 17), demonstrating that it generalizes beyond SP data and does not overfit to SP-specific patterns.
>
> **Choice of grouping features and number of groups**. We have added a detailed sensitivity analysis in Appendix A.9 evaluating six different feature sets—both importance-guided and randomly selected. The results show a clear trade-off: using many grouping features creates too many small, highly imbalanced groups, while using too few features collapses distinct behavioural modes. The final feature set (Age_group, Beginners, NY_CA, Part) produces the best balance between (i) capturing behavioural heterogeneity and (ii) maintaining stable group sizes and reasonable acceptance-rate distributions. This analysis justifies why we use four features and why the resulting grouping structure (32 groups for ~3,000 samples) is appropriate and empirically validated. For your convenience, we present the results below for your reference:
>
> ### Summary of persona grouping experiments across six feature sets
>
> | Feature Set | #Groups | Var(p_accept) | #Imbalanced Groups | Imbalance Score |
> |-------------|---------|----------------|----------------------|------------------|
> | F1 | 117 | 0.0304 | 18 | 1.0000 |
> | F2 | 63  | 0.0281 | 5  | 0.9661 |
> | F3 | 33  | 0.0273 | 5  | 0.7419 |
> | F4 | 18  | 0.0226 | 2  | 0.5396 |
> | F5 | 10  | 0.0248 | 0  | 0.5689 |
> | F6 | 32  | 0.0194 | 1  | 0.5417 |
>
> **Feature sets:**
> - **F1** = {Age_group, Degree, Acceptance, Peak, Gender, Morning}
> - **F2** = {Age_group, Degree, Acceptance, Peak, Gender}
> - **F3** = {Age_group, Acceptance, Beginners, Degree}
> - **F4** = {Age_group, Acceptance, Beginners}
> - **F5** = {Age_group, Acceptance}
> - **F6** = {Age_group, Beginners, NY_CA, Part}
>
> **Scale of generated data**. We generate 3,000 synthetic samples to match the scale of the SP dataset, ensuring balanced contribution without overwhelming real data. Larger-scale generation is possible, but our experiments show that the chosen size already provides substantial and stable performance gains.
>
> > Q1. In Figure 1, 20% training was used for evaluation, do you mean test set here?
>
> A: Thank you for pointing that out. The 20% refers to the testing set and has been corrected.

---

### Meta-Review · Area_Chair_zRrC · 2026-01-11

**Summary:**

This paper introduces LLM-OAP, a framework aimed at enhancing the prediction of order acceptance behavior among ride-hailing drivers through synthetic data generation using Large Language Models (LLMs). The methodology includes feature-aware persona grouping, LLM-based data augmentation, and confidence-based curation. While the approach demonstrates marginal improvements in accuracy over baseline models on stated-preference (SP) survey data, several critical limitations hinder its overall impact and reproducibility.

Firstly, the novelty of the core approach is limited, as it mainly integrates existing techniques without introducing fundamental innovations. Reproducibility is another major issue, with incomplete reporting of hyperparameters and filtering statistics, making replication of results challenging. Furthermore, the paper does not provide any mechanistic understanding of why the augmentation improves model performance or address issues related to LLM hallucinations and adversarial robustness.

**Reviewer Scores:**

NA

---

### Decision · Program_Chairs · 2026-01-26

Reject